# Zinc shapes the folding landscape of p53 and establishes a pathway for reactivating structurally diverse cancer mutants

Adam R Blanden[1†], Xin Yu[2†], Alan J Blayney[3], Christopher Demas[3], Jeung-Hoi Ha[3], Yue Liu[2], Tracy Withers[2], Darren R Carpizo[4‡*], Stewart N Loh[3‡*]

[1]Department of Neurology, SUNY Upstate Medical University, Syracuse, Syracuse, United States; [2]Rutgers Cancer Institute of New Jersey, Department of Surgery, Rutgers Robert Wood Johnson Medical School, New Brunswick, United States; [3]Department of Biochemistry and Molecular Biology, SUNY Upstate Medical University, Syracuse, United States; [4]Department of Surgery, University of Rochester School of Medicine and Dentistry and Wilmot Cancer Center, Rochester, United States

**Abstract** Missense mutations in the p53 DNA-binding domain (DBD) contribute to half of new cancer cases annually. Here we present a thermodynamic model that quantifies and links the major pathways by which mutations inactivate p53. We find that DBD possesses two unusual properties—one of the highest zinc affinities of any eukaryotic protein and extreme instability in the absence of zinc—which are predicted to poise p53 on the cusp of folding/unfolding in the cell, with a major determinant being available zinc concentration. We analyze the 20 most common tumorigenic p53 mutations and find that 80% impair zinc affinity, thermodynamic stability, or both. Biophysical, cell-based, and murine xenograft experiments demonstrate that a synthetic zinc metallochaperone rescues not only mutations that decrease zinc affinity, but also mutations that destabilize DBD without impairing zinc binding. The results suggest that zinc metallochaperones have the capability to treat 120,500 patients annually in the U.S.

**\*For correspondence:**
darren_carpizo@urmc.rochester.edu (DRC);
lohs@upstate.edu (SNL)

†These authors contributed equally to this work
‡These authors also contributed equally to this work

**Competing interests:** The authors declare that no competing interests exist.

## Introduction

The transcription factor p53 regulates a host of cellular responses to damage and distress (*Kruiswijk et al., 2015*). Its abilities to halt cell cycle progression, upregulate DNA repair pathways, and induce apoptosis help prevent deleterious mutations from propagating in cell populations. Mutations in p53 are an established driver of human cancer (*Bouaoun et al., 2016*). The mutational spectrum of p53 is atypical because tumorigenic alterations are overwhelmingly missense and map to nearly every position within one of the domains of the protein (the DNA-binding domain, or DBD) (*Baugh et al., 2018*). By contrast, other frequently-mutated tumor suppressors such as BRCA1/2 (*ARUP Scientific Resource, 2020*), retinoblastoma 1 (*LOVD Gene homepage, 2018*), and *PTEN, 2020* are found with mostly nonsense, deletion, or insertion mutations. From a structural standpoint, p53 DBD is unusual in that it is marked by low thermodynamic and kinetic stability (*Joerger and Fersht, 2010*). The apparent melting temperature ($T_m$) of wild-type (WT) DBD has been measured to be 32–45°C depending on buffer composition (*Bell et al., 2002*; *Butler and Loh, 2006*; *Friedler et al., 2003*), and the protein unfolds with a half-time of 9 m at 37°C (*Friedler et al., 2003*). As a result of this borderline stability, many tumorigenic mutations reduce $T_m$ below body

**Figure 1.** X-ray structure of WT DBD showing locations of mutations characterized in this study. Alpha carbons of mutated residues are colored according to their classifications described in *Results*: zinc-binding class (green), stability class (red), DNA-binding class (blue), and mixed zinc-binding/stability class (orange). DNA and $Zn^{2+}$ are the gray surface and black sphere, respectively. PDB 1TSR.

temperature and/or increase the rate at which the protein unfolds (*Butler and Loh, 2006*; *Friedler et al., 2003*).

On the basis of p53's instability at physiologic conditions is its interaction with zinc. p53 consists of the N-terminal transactivation domain, a central DBD and the C-terminal tetramerization domain. The X-ray crystal structure of DBD (residues 94–312) reveals a β-sandwich with a DNA-binding surface consisting of a loop-sheet-helix motif and two loops (L2 and L3) (*Figure 1*; *Cho et al., 1994*). These loops are stabilized by the tetrahedral coordination of a single zinc ion by C176 and H179 of L2 and C238 and C242 of L3. Removing $Zn^{2+}$ from DBD causes loss of DNA-binding specificity,

widespread changes in the protein NMR spectrum, and a reduction of ~3 kcal mol$^{-1}$ in the apparent folding free energy (*Butler and Loh, 2003*). This inherent malleability has also been demonstrated in cells by overexpressing $Zn^{2+}$-chelating proteins (metallothioneins) or adding small-molecule $Zn^{2+}$ chelators, and observing reversible loss of sequence-specific DNA-binding activity and a switch in recognition by an antibody that recognizes native p53 (PAB1620) to one that binds to unfolded/mis-folded p53 (PAB240) (*Méplan et al., 2000*).

Many tumorigenic mutations occur in and around the zinc-binding site and presumably impair this interaction. The most common p53 mutation in cancer, R175H, is immediately adjacent to the zinc-chelating residue C176 (*Figure 1*) and dramatically weakens metal-binding affinity (*Butler and Loh, 2003*; *Yu et al., 2014*). Furthermore, there is evidence that some cancers inactivate p53 by upregulating metallothioneins and starving wild-type (WT) p53 of zinc (*Mehrian-Shai et al., 2015*). These observations have led our group and others to develop a new class of p53-targeted therapeutics based on $Zn^{2+}$ delivery. By our definition, zinc metallochaperones (ZMCs) reactivate mutant p53 by shuttling $Zn^{2+}$ from extracellular sources through the plasma membrane and into cells, thereby increasing intracellular $Zn^{2+}$ concentrations to levels high enough to remetallate mutant p53. This approach has proven effective for multiple mutants in cell culture and mouse models of cancer (*Yu et al., 2014*; *Blanden et al., 2015*; *Garufi et al., 2013*).

Although the importance of zinc to p53 structure/function has long been recognized, there is still no quantitative thermodynamic model describing the p53-$Zn^{2+}$ interaction and its linkage to folding. Previous attempts to characterize DBD folding thermodynamics have yielded valuable insight into mechanisms of p53 dysfunction, and have informed drug development efforts, but are ultimately incomplete because they lack information regarding zinc-binding affinity of folded and unfolded states (*Bullock et al., 2000*; *Bullock et al., 1997*). Furthermore, there is disagreement in the literature regarding which mutants are potentially treatable using zinc-based therapies, likely because of the difficulty of inferring physical mechanisms from complex biological data and lack of a consensus definition regarding what constitutes a zinc-binding mutant (*Garufi et al., 2013*; *Salim et al., 2016*).

Here, we present and validate a thermodynamic model that partitions DBD-folding energy into two measurable quantities: the free energy of folding of zinc-free DBD (apoDBD; $\Delta G_{apo}$), and the free energy of $Zn^{2+}$ binding ($\Delta G_{Zn}$) to both native and non-native sites. The data reveal for the first time that: (i) $\Delta G_{apo}$ is extremely unfavorable at 37°C, indicating that wild-type (WT) DBD is intrinsically unfolded in the absence of zinc, and (ii) DBD has one of the highest zinc-binding affinities of any eukaryotic protein yet reported (*Kochańczyk et al., 2015*). Remarkably, the unusually large unfavorable value of $\Delta G_{apo}$ and the atypically large favorable value of $\Delta G_{Zn}$ are predicted to nearly cancel each other out at physiological temperature and intracellular zinc concentration, causing the overall free energy change of folding to be near zero.

Although p53's instability at 37°C has been previously documented, our modeling emphasizes that p53 is poised between folded and unfolded conformations in the cell, with available zinc concentration being a major determining factor. It further holds that mutations that decrease protein stability (but not zinc-binding affinity) and mutations that decrease zinc-binding affinity (but not protein stability) both cause p53 to lose function by a common unfolding mechanism, and both might be similarly rescued by increasing the concentration of available $Zn^{2+}$. As a test, we apply the model to 22 of the most prevalent cancer-associated p53 variants and classify them into three classes—stability, zinc-binding, or DNA-binding—based on (respectively) $\Delta G_{apo}$, $K_{Zn}$, and $K_{DNA}$, the latter being dissociation constants for binding to a panel of p53 recognition elements. We validate the model by testing whether ZMC1 reactivates members of the stability class of p53 mutants in cells, a category not previously regarded as being amenable to $Zn^{2+}$ therapy. The results provide a more complete picture of the p53 activation/inactivation landscape and significantly expand the number of p53 mutants that are potentially rescuable by ZMCs.

## Results

### DBD-zinc energy landscape

First, we developed a thermodynamic model to describe the conformational states of DBD as a function of [$Zn^{2+}$]. We propose a minimal four-state mechanism in which $Zn^{2+}$ binds to a single, high-affinity site in native apoDBD (N) with an equilibrium constant of $1/K_{Zn}$ (*Equation 1*) and to one or

more low-affinity sites in unfolded apoDBD (U) with an average equilibrium constant of $1/K_{Zn,U}$ (*Equation 2*). HoloDBD (N·Zn) is the only species with native DNA-binding activity.

$$N + Zn \rightleftharpoons N \cdot Zn; \frac{1}{K_{Zn}} \tag{1}$$

$$U + Zn \rightleftharpoons U \cdot Zn; \frac{1}{K_{Zn,U}} \tag{2}$$

To measure $K_{Zn}$, we monitored the increase in Tyr fluorescence (*Butler and Loh, 2003*) of apoDBD as it binds $Zn^{2+}$ at 10°C (*Figure 2A*). $[Zn^{2+}]_{free}$ was buffered at the indicated concentrations using chelators of varying $Zn^{2+}$ affinity. Fitting the data to the one-site binding equation yields $K_{Zn}$ = $(1.6 \pm 0.3) \times 10^{-15}$ M. To our knowledge, this is one of the lowest $K_{Zn}$ values ever reported, with only one other eukaryotic protein possessing comparable affinity (PDZ and LIM domain protein 1; $K_{Zn}$ = $3.2 \times 10^{-15}$ M) (*Kochańczyk et al., 2015*). Because p53 is a tetramer, we measured $K_{Zn}$ of the full-length protein to determine if there is cooperativity between the monomers (*Figure 2A*). Fitting the data to the Hill-binding equation reveals $K_{Zn}$ = $(0.4 \pm 0.1) \times 10^{-15}$ M and n = $0.99 \pm 0.02$, indicating that each monomer binds zinc independently and that the isolated DBD is an accurate representation of the DBD in the full-length p53 tetramer. We then measured $K_{Zn,U}$ by competition assay between urea-denatured apoDBD and the fluorescent $Zn^{2+}$ chelator FluoZin-3 (*Figure 2B*). The apparent $K_{Zn,U}$ value of $(42 \pm 7) \times 10^{-9}$ M is $10^7$-fold weaker than $K_{Zn}$. We generated a Ser mutant of the zinc-binding residue C176 and found its $K_{Zn,U}$ to be essentially identical ($(50 \pm 23) \times 10^{-9}$ M), indicating that the zinc-binding sites in the native and unfolded proteins are distinct (*Figure 2B*).

To delineate the relationship between zinc binding and WT DBD folding, we performed urea denaturation experiments at various concentrations of $[Zn^{2+}]_{free}$ that were fixed by a mixture of $Zn^{2+}$ chelators (*Figure 2—figure supplement 1A*). The overall free energy change for folding to either apo or holo native states ($\Delta G_{fold}$) is given by (*Equation 3*; *Pace and McGrath, 1980*), where $\Delta G_{Zn}$ = $-RT \cdot \ln(1+K_{Zn}^{-1}[Zn^{2+}]_{free})$, $\Delta G_{Zn,U}$ = $-RT \cdot \ln(1+K_{Zn,U}^{-1}[Zn^{2+}]_{free})$, and $\Delta G_{apo}$ = $-RT \cdot \ln(K_{apo})$ (*Equation 4*).

$$\Delta G_{fold} \rightleftharpoons \Delta G_{apo} - \Delta G_{Zn} + \Delta G_{Zn,U} \tag{3}$$

$$U \rightleftharpoons N; K_{apo} \tag{4}$$

Zinc-induced stabilization can be visualized intuitively by plotting $\Delta G_{fold}$ against $[Zn^{2+}]_{free}$ (*Figure 2C*). Focusing on the WT DBD data, the curve begins as a horizontal line that intersects the y-axis at $\Delta G_{apo}$, then starts to deflect upward when zinc begins to bind apoDBD, i.e. when $[Zn^{2+}]_{free} \approx K_{Zn}$. $\Delta G_{fold}$ continues to increase linearly with $\log[Zn^{2+}]_{free}$ and only levels off when zinc starts to bind unfolded DBD ($[Zn^{2+}]_{free} \approx K_{Zn,U}$). A pure zinc-binding mutant such as R175H weakens $\Delta G_{Zn}$ (*Equation 1*) without affecting $\Delta G_{apo}$, shifting the deflection point to higher $[Zn^{2+}]_{free}$ but maintaining the same y-intercept as WT (*Figure 2C*). Conversely, the signature of pure stability mutants such as A138V, a well-established temperature-sensitive variant (*Cuddihy et al., 2008*) is a similar deflection point compared to WT but a y-intercept value closer to zero. We observed no trend in the cooperativity parameter (*m*-value) of the unfolding transitions over $10^{-16}$ M < $[Zn^{2+}]_{free}$ <$10^{-12}$ M (*Figure 2—figure supplement 1A*), signifying that urea denaturation is adequately described by a two-state transition between unfolded and native apo or native holo states (the *m*-values of the two are similar [*Butler and Loh, 2003*]) throughout the tested zinc concentration. We also performed denaturation experiments starting with either apoDBD or zinc-bound DBD and obtained similar results, indicating that metal binding had reached equilibrium during the incubation time (*Figure 2—figure supplement 1B*). The data are described well by *Equation 3* and yield fit parameters of $\Delta G_{apo}$ = $6.4 \pm 0.1$ kcal mol$^{-1}$ and $K_{Zn}$ = $(7.0 \pm 2.5) \times 10^{-15}$ M, both of which are in good agreement with direct measurements (*Figure 3—source data 1*).

## Extrapolation to physiological conditions

Because our DBD experiments must be performed at low temperature to avoid protein aggregation, we sought to gain insight into how $\Delta G_{apo}$ changes with temperature. To quantify this, we performed urea melts of apoDBD at 10 temperatures over the range 2–22°C. The resulting curve is a parabola

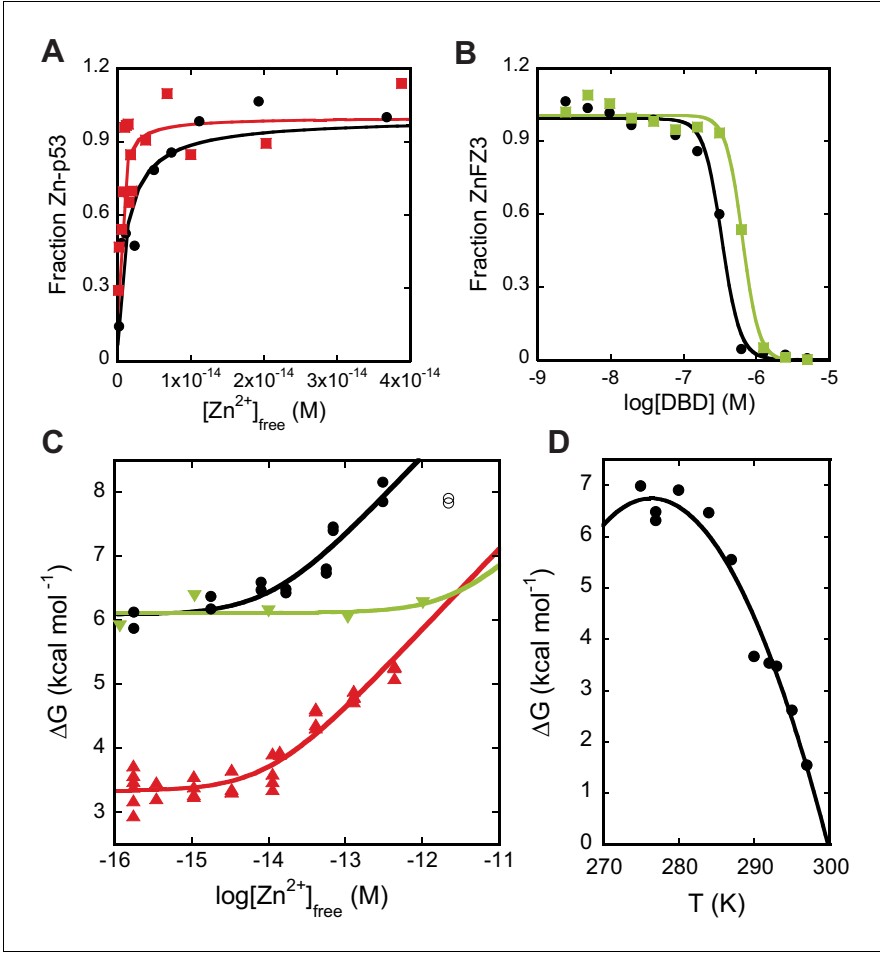

**Figure 2.** Zinc-binding affinity and stability of DBD. (A) WT DBD (black) and full-length WT p53 (red) bind $Zn^{2+}$ with $K_{Zn}$ values of (1.6 ± 0.3) x $10^{-15}$ M and (0.4 ± 0.1) x $10^{-15}$ M, respectively, as determined by change in Tyr fluorescence (10°C, n = 3, SD). (B) Unfolded WT DBD (black) and unfolded C176S DBD (green), bind $Zn^{2+}$ with $K_{Zn, U}$ values of (42 ± 7) x $10^{-9}$ M and (50 ± 23) x $10^{-9}$ M, respectively, as determined by FluoZin-3 competition in 6 M urea (10°C, n = 3, SD). (C) Plotting folding free energy of DBD vs. $[Zn^{2+}]_{free}$ (10°C) reveals that R175H (green) is a pure zinc-binding-class mutant whereas A138V (red) is a pure stability-class mutant. The point at which the lines deflect upwards are the approximate $K_{Zn}$ values. WT DBD is in black. Open points denote outliers excluded from analysis. Outliers were identified on inspection and rejected if their exclusion (1) improved goodness of fit, and (2) produced a model for which they lay outside the 95% prediction interval. (D) Temperature dependence of apoDBD folding free energy fit to the Gibbs-Helmholtz equation (*Figure 2—figure supplement 1C*) yields $\Delta H_m$ = 171 ± 20 kcal $mol^{-1}$, $T_m$ = 300 ± 1 K, and $\Delta C_p$ = 7.0 ± 1.7 kcal $mol^{-1}$ $K^{-1}$ (fit value ± SE of fit). In (C and D), independent experimental data were pooled and fit once, and results are reported as the fit parameters and standard effort of the fit. Otherwise (A, B), replicates consisted of independent experiments performed with the same preparations of purified proteins, which were fit separately and the results pooled. Single curves are shown in the figure for illustration.

The online version of this article includes the following figure supplement(s) for figure 2:

**Figure supplement 1.** Physical analysis validation data.

the narrowness of which is proportional to $\Delta C_p$, the change in heat capacity on protein unfolding at constant pressure (*Figure 2D*). We fit these data to the Gibbs-Helmholtz equation (*Figure 2—figure supplement 1C*) to obtain $\Delta C_p$, melting temperature ($T_m$), and the enthalpy change of unfolding at $T_m$ ($\Delta H_m$). Introduced by *Pace and Laurents, 1989*, this method has been shown to reproduce $\Delta C_p$, $T_m$, and $\Delta H_m$ obtained by scanning calorimetry for a number of proteins (*Pace et al., 1999*; *Talla-Singh and Stites, 2008*). We then combined the full four-state model (*Equations 1, 2, 4*) with the Gibbs-Helmholtz equation to project the energy landscape of WT DBD as a three-dimensional plot

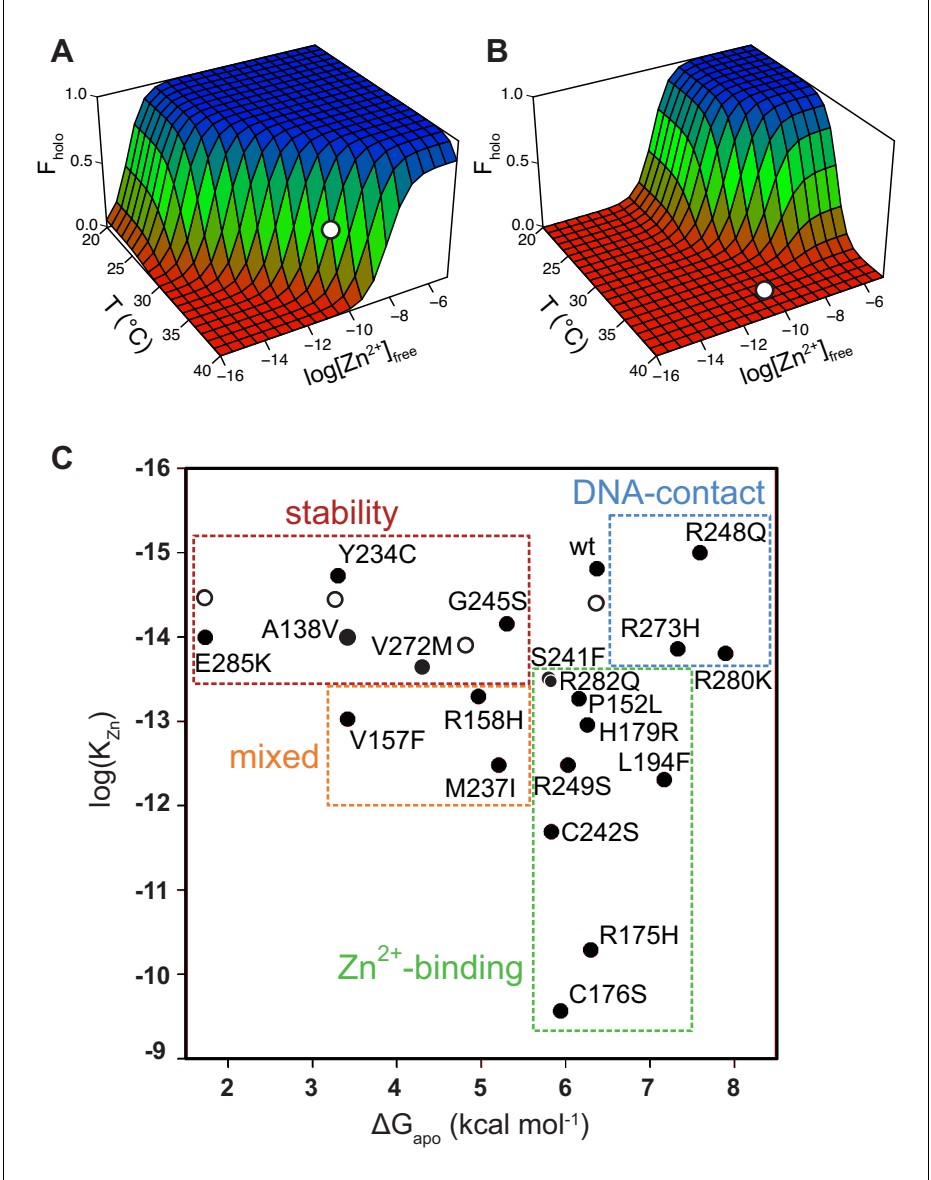

**Figure 3.** Energy landscape of DBD folding and classification of p53 mutants. The populations of folded, active WT DBD (**A**) and R175H DBD (**B**) depend strongly on free zinc concentration and temperature. White circles indicate physiological T and $[Zn^{2+}]_{free}$. (**C**) 17 of the top 20 most common tumorigenic p53 mutations impair DBD thermodynamic stability (red box), zinc-binding affinity (green box), or both (orange box) (10°C). The remaining three are DNA-contact mutations (blue box). Open circles indicate WT DBD destabilized by urea.

The online version of this article includes the following source data and figure supplement(s) for figure 3:

**Source data 1.** Stabilities and zinc-binding affinities of apoDBD variants (10°C).

**Figure supplement 1.** Free energy of folding versus buffered $[Zn^{2+}]_{free}$ for p53 DBD mutants.

**Figure supplement 2.** Free energy of folding versus buffered $[Zn^{2+}]_{free}$ for p53 DBD mutants.

---

with $[Zn^{2+}]_{free}$, temperature, and fraction holoDBD ($F_{holo}$) on the x-, y-, and z-axes, respectively (**Figure 3A**).

When the protein is kept cool (10°C) it is stable even without zinc ($\Delta G_{apo} = -6.3$ kcal mol$^{-1}$; **Figure 2D**), making $F_{holo}$ effectively unity at all relevant concentrations of free zinc ($>10^{-15}$ M; **Figure 3A**). At 37°C, however, the protein is completely unfolded without zinc ($\Delta G_{apo} = 6.9$ kcal mol$^{-1}$). Strikingly, the high zinc-binding affinity of WT apoDBD together with the concentration of

free zinc in the typical cell ($10^{-10}$ M) (*Krezel and Maret, 2006*; *Vinkenborg et al., 2009*) result in an overall $\Delta G_{fold}$ near zero and $F_{holo}$ = 0.43 (*Figure 3A*). This suggests that under normal physiological conditions, WT p53 is balanced on the edge of folding and unfolding and may be pushed in either direction by surprisingly small changes in $[Zn^{2+}]_{free}$, $\Delta G_{apo}$, $K_{Zn}$, or temperature. For example, raising $[Zn^{2+}]_{free}$ to $10^{-9}$ M or lowering it to $10^{-11}$ M yields $F_{holo}$ values of 0.9 and 0.08, respectively. $F_{holo}$ changes similarly when $K_{Zn}$ is altered by a factor of 10, $\Delta G_{apo}$ by 1–2 kcal mol$^{-1}$, and temperature by 2°C.

The conclusion that WT p53 folding is sensitive to $[Zn^{2+}]_{free}$ fluctuations in the physiological regime carries significant implications, and uncertainty of the $\Delta G_{apo}$ extrapolation as well as technical artifacts must be considered. The accuracy of the extrapolation and of our zinc-binding model as a whole can be evaluated by comparing our results to those of Fersht and coworkers, who applied the same Gibbs-Helmholtz analysis to holoDBD stability over a similar temperature range (5–25°C) (*Bullock et al., 2000*). To connect the two data sets, we calculated $F_{holo}$ using our energy landscape model with $[DBD]_{total}$ and $[Zn^{2+}]_{total}$ set to those used in the Fersht group's experiments (2.5 μM each). We then obtained the theoretical stability of holoDBD ($\Delta G_{holo,theory}$) using *Equation 5* and compared this to the value determined by Fersht et al. ($\Delta G_{holo}$).

$$\Delta G_{holo,theory} = -RT \cdot ln\frac{F_{holo}}{1 - F_{holo}} \tag{5}$$

At 10°C, where $\Delta G_{apo}$ and $\Delta G_{holo}$ are both determined experimentally, $\Delta G_{holo,theory}$ (−12.3 kcal mol$^{-1}$) and $\Delta G_{holo}$ (−10.6 kcal mol$^{-1}$) are in good agreement. At 37°C, where $\Delta G_{apo}$ and $\Delta G_{holo}$ are extrapolated values, $\Delta G_{holo,theory}$ and $\Delta G_{holo}$ are nearly identical (−2.9 kcal mol$^{-1}$ and −3.0 kcal mol$^{-1}$, respectively). Given DBD's high-affinity for zinc ($K_{Zn}$ = 1.6 fM), the only way to obtain $\Delta G_{holo}$ = −3.0 kcal mol$^{-1}$ at 37°C and 2.5 μM total zinc is for apoDBD to be as unstable as the landscape model predicts.

We note, however, that the $\Delta C_p$ value we obtain is substantially higher than the figure calculated based on the change in accessible surface area of holoDBD upon unfolding (3.8 kcal mol$^{-1}$ K$^{-1}$) (*Bullock et al., 2000*; *Myers et al., 1995*). To eliminate the possibility of buffer or denaturant-specific effects, we changed the buffer from Tris to phosphate and the denaturant from urea to guanidine hydrochloride and obtained nearly identical results (*Figure 2—figure supplement 1E*). There was no trend in *m*-value that would signify deviation from two-state behavior (*Figure 2—figure supplement 1D*). We therefore conclude that apoDBD, owing to its large $\Delta C_p$ value, possesses a stability that is unusually dependent on temperature.

The energy landscape of mutant DBD can be generated in the same manner as that of WT DBD provided that the mutation does not alter the enthalpy change of unfolding. This scenario is reasonable for the zinc-binding class, because these mutations do not perturb $\Delta G_{apo}$ and thus $\Delta H$ and $\Delta S$ can be assumed to remain unchanged. Stability-class mutations, however, can affect $\Delta H$, $\Delta S$, or both, making the extrapolation of $\Delta G_{fold}$ to 37°C unreliable.

## Categorization of p53 mutants

We then sought to apply this methodology to gain insight into the mechanisms by which tumorigenic mutations cause p53 to lose function, we purified 22 DBD missense mutants commonly found in human cancer. We chose the most frequent somatic mutations in the IARC database with the following additional criteria: (i) when multiple mutations were reported at a single position, only the most common variant was used (to maximize protein coverage); (ii) if the residue was mutated to Trp, we selected the next most frequent alteration at that position (an extra Trp interferes with our fluorescence assays); (iii) when multiple mutations of a zinc-coordinating residue were common, we chose the most isosteric in order to help isolate the effects of $Zn^{2+}$ binding. We performed urea denaturation experiments as a function of buffered $[Zn^{2+}]_{free}$ to determine $\Delta G_{apo}$ and $K_{Zn}$ at 10°C (*Figure 3—figure supplement 1* and *Figure 3—figure supplement 2*), and we also cross-checked $K_{Zn}$ using direct $Zn^{2+}$ binding experiments (c.f. *Figure 2A*) for select mutants. We were able to determine $\Delta G_{apo}$ for all mutants and $K_{Zn}$ for all but three (Y205C, Y220C, Y163C; *vide infra*) (*Figure 3—source data 1*).

By placing the 22 variants on a plot according to their $\Delta G_{apo}$ and $K_{Zn}$ values, three classes of mutants emerge: pure zinc-binding, pure stability, and DNA-binding, with a sub-class of mixed zinc-

binding/stability phenotype (*Figure 3C*). Pure zinc-binding mutants exhibit decreased $Zn^{2+}$ affinity but normal stability and thus cluster near a vertical line drawn directly below WT DBD (green box). Members of this class include R175H as well as the direct $Zn^{2+}$ ligating mutants C176S, C242S, and H179R. Significantly, the analysis reveals a number zinc-binding mutations (P152L, L194F, R282Q) that were not previously suspected as being so due to their long distances from the metal-binding pocket (*Figure 1*).

Pure stability mutants are less stable than WT but suffer no zinc-binding deficiency; these are defined by falling near the horizontal line drawn to the left of WT DBD (red box in *Figure 3C*). To validate that protein stability can indeed be separated from zinc affinity in this manner, we destabilized WT apoDBD using sub-denaturing amounts of urea and measured $K_{Zn}$ using the Tyr fluorescence assay. These data (open circles) fall on the horizontal line as predicted by the model. The mixed-phenotype mutants (A138V, V267F, R158H, M237I; orange box in *Figure 3C*) decrease both stability and zinc affinity. Only three mutations (R248Q, R273H, R280K) do not impair stability or metal affinity (blue box in *Figure 3C*). These residues are in direct contact with DNA (*Figure 1*). DNA-contact mutants are more stable than WT, likely because they remove or reposition a positive charge and reduce repulsions in the highly cationic DNA-binding groove (*Butler and Loh, 2003*; *Bullock et al., 2000*). Our data confirm that they bind zinc normally and have lost DNA-binding affinity (*vide infra*).

*Figure 3C* reveals a remarkable range in magnitude of the stability and zinc-binding deficiencies. The most severe member of each class reduces $\Delta G_{apo}$ by 4.6 kcal $mol^{-1}$ (E285K) and $K_{Zn}$ by 5 orders of magnitude (C176S). Additionally, several zinc-binding or mixed mutations (L194F, R282Q, V157F, R158H, P152L) lie far from the canonical zinc-binding pocket, suggesting long-range communication between distant regions of DBD. The subtle nature of DBD structure and energetics is also evident from the observation that adjacent mutations can produce markedly different effects. For example, R248Q is a classic DNA-contact mutant whereas R249S is a pure zinc-binding mutant. This demonstrates the difficulty in inferring physical consequences of p53 mutation based on location and stresses the need to measure properties of each mutant to gain a clear understanding of its impairment.

## DNA-binding affinity

To gain a better understanding of the effects of p53 mutation on DNA-binding activity, we measured dissociation constants ($K_{DNA}$) for binding of the 22 DBD mutants to a panel of fluorescently labeled oligonucleotides bearing 10 p53 recognition elements (p53RE; *Supplementary file 1A*). The fluorescence anisotropy data fit adequately to the Hill-binding equation. When global fits for individual sequences were performed by linking the Hill parameter and curve amplitudes, the resultant curves fit equally well regardless of DBD mutant, indicating similar binding mechanisms regardless

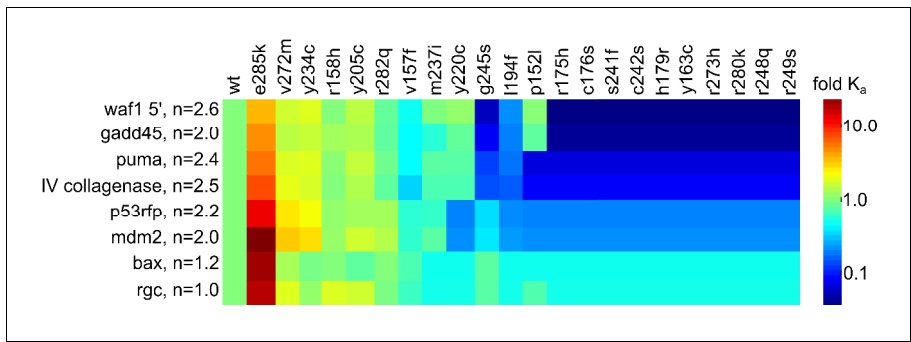

**Figure 4.** Heatmap of DNA-binding of p53 DBD mutants. Affinity of mutant DBD for DNA oligonucleotides, expressed relative to WT DBD (4°C). Oligonucleotide names and Hill coefficients (**n**) are on the left. DNA sequences are in *Supplementary file 1A*.

The online version of this article includes the following source data and figure supplement(s) for figure 4:

**Source data 1.** $K_{DNA}$ values used to generate the heatmap in *Figure 4*.

**Figure supplement 1.** DNA-binding of p53 DBD mutants.

of mutation (*Figure 4—figure supplement 1A*). *Figure 4* represents the $K_{DNA}$ values relative to that of WT DBD in the form of a heat map. We excluded the two p53REs to which none of the DBD variants bound (WAF1-3' and EGFR), and assigned $K_{DNA}$ = 25 µM (the weakest value we could consistently measure) to data sets in which we could detect no interaction (*Figure 4—source data 1*).

By visual inspection, most of the stability and mixed mutants maintain similar binding affinity for the different p53REs, whereas all of the DNA-contact mutants and most of the zinc-binding mutants lose measurable affinity for all p53REs (*Figure 4*). The majority of zinc-binding mutants purify with sub-stoichiometric but detectable zinc content (e.g. R175H contains 0.6 equivalents of $Zn^{2+}$ [14]), and consequently should show partial DNA-binding activity. We speculated that the loss of DNA-binding may be caused by $Zn^{2+}$ misligating to non-native sites ($K_{Zn,U}$ = 42 x $10^{-9}$ M; *Figure 2B*) that outcompete the native site at the temperature of protein expression (18°C). To test this hypothesis, we attempted to re-establish native metallation status to the zinc-binding mutants P152L, R175H, H179R, and L194F by first removing all bound metal and then remetallating using a EGTA/ZnCl$_2$ buffering system. This procedure restored measurable DNA-binding affinity (WAF1 oligonucleotide) to all four mutants (*Figure 4—figure supplement 1B*). The positive control (WT) and negative control (R280K) yielded the expected results of normal and undetected DNA-binding, respectively. These data suggest that zinc misligation contributes to loss of DNA-binding activity for this class of mutants.

Interestingly, while most of the data are explained by global increases and decreases in affinity based on thermodynamic category, there do seem to be mutation-specific effects as well. P152L maintains WT-like affinity for WAF1 5', GADD45, and RGC, but loses all measurable affinity for PUMA, Type IV collagenase, p53RFP, MDM2, and BAX. As another example from the stability mutant category, Y220C maintains affinity for WAF1 5', GADD45, PUMA, and Type IV Collagenase, but seems to lose affinity for p53RFP, MDM2, BAX, and RGC. Additionally, E285K gains affinity for all p53REs, but the increase in affinity ranges from threefold for WAF1 5' to 15-fold for MDM2. E285K may bind more tightly to all DNA sequences due to enhanced electrostatic interactions with the phosphate backbone afforded by the negative-to-positive charge reversal near the active site. These results indicate that there are conserved functional consequences of the thermodynamic impairments we measure for p53 mutants that account for the majority of the functional differences we see in DNA-binding phenotype, but there are also mutation-specific effects that are better explained by idiosyncratic structural changes caused by each mutant that our model does not capture.

## Tyr to Cys mutants are exceptions to the model

Y163C, Y205C, Y220C, and Y234C are all destabilized ($\Delta G_{apo}$ = 3.5–5.0 kcal/mol), but, unlike all other members of the stability class except for M237I, their $\Delta G_{fold}$ values fail to increase with $[Zn^{2+}]_{free}$ (*Figure 3—figure supplement 1*). The flat profiles of these plots can potentially be explained by very weak $K_{Zn}$ values, but two observations argue against this interpretation. First, direct measurement of $K_{Zn}$ by Tyr fluorescence yields a value of (1.9 ± 0.2) x $10^{-15}$ M for Y234C (*Figure 3—source data 1*), the only Tyr→Cys mutant that was able to survive the zinc removal procedure without aggregating. Second, Y205C, Y220C, and Y234C retain WT-like DNA-binding activity, whereas most of the zinc-binding mutants are compromised in this regard. We hypothesized that the extra Cys residue forms a new $Zn^{2+}$ interaction site in the unfolded state, thereby bringing $K_{Zn,U}$ closer to $K_{Zn}$ and siphoning the native state of the stabilization energy it would normally receive by binding $Zn^{2+}$. To test this hypothesis, we denatured Y234C apoDBD in 6 M urea and measured $K_{Zn,U}$ by Tyr fluorescence. Unfolded Y234C binds zinc twice as tightly as WT [$K_{Zn,U}$ = (20 ± 4) x $10^{-9}$ M; n = 6], consistent with the hypothesis with the caveat that the unfolded state in urea is likely different from that in buffer. As an additional test, we made the Y234A mutant and found that this mutation restores the relationship between $\Delta G_{fold}$ and $[Zn^{2+}]_{free}$ as predicted by the model (*Figure 3—figure supplement 2*), with a $K_{Zn}$ value identical to that of WT within experimental error (*Figure 3—source data 1*). These results demonstrate that Y234A is a stability-class mutant and suggest that other Tyr→Cys mutants may also be members of this category. The data also indicate that introducing Cys can encourage zinc misligation in the unfolded state and destabilize native p53 through manipulation of $K_{Zn,U}$ in our model.

## Testing the energy landscape model in cells using ZMC1

We previously demonstrated that the small-molecule zinc metallochaperone ZMC1 can reactivate p53 mutants of the zinc-binding class in cultured cells. ZMC1 forms a 2:1 complex with $Zn^{2+}$ and acts as an ionophore to transport $Zn^{2+}$ into the cell, whereupon it buffers $[Zn^{2+}]_{free}$ to 10–20 nM (*Yu et al., 2014*). When ZMC1 was added to human cancer cell lines homozygous for p53[R175H] or one of the direct zinc ligation mutations (p53[C176F], p53[C238S], p53[C242S]), cell toxicity was observed with $EC_{50}$ values well below that of p53[WT] and p53-null controls. This enhanced ZMC1 sensitivity was shown to be due to a p53-mediated apoptotic program (*Yu et al., 2012*). Given the ZMC mechanism, we previously surmised that the spectrum of mutants to which ZMCs were amenable was limited to those with impaired zinc binding. However, the relationship of zinc binding to the energy of protein folding we observed in our model allowed us to hypothesize that raising intracellular zinc concentrations could increase protein stability enough to rescue wild-type conformation of some stability mutants. To test this, we used ZMC1 as a tool to raise intracellular concentrations of zinc in an array of zinc-binding, stability, and mixed classes of p53 mutants generated by site specific mutation in plasmids and expressed in H1299 (p53-null) cells. We also used human tumor cell lines that endogenously express mutant p53, when available. Cell survival curves are shown in *Figure 5* and ZMC1 $EC_{50}$ values are summarized in *Table 1*.

The mutants tested in the pure zinc-binding class are R175H, C176S, L194F, P152L, and R282Q. Cells transfected with R175H, C176S, and L194F show >100 fold enhanced sensitivity to ZMC1 ($EC_{50}$ = 0.002 μM – 0.005 μM) relative to untransfected and vector-only controls (*Table 1*). A p53[L194F] tumor cell line is also sensitive to ZMC1 (*Figure 7—figure supplement 1A*). $EC_{50}$ values of

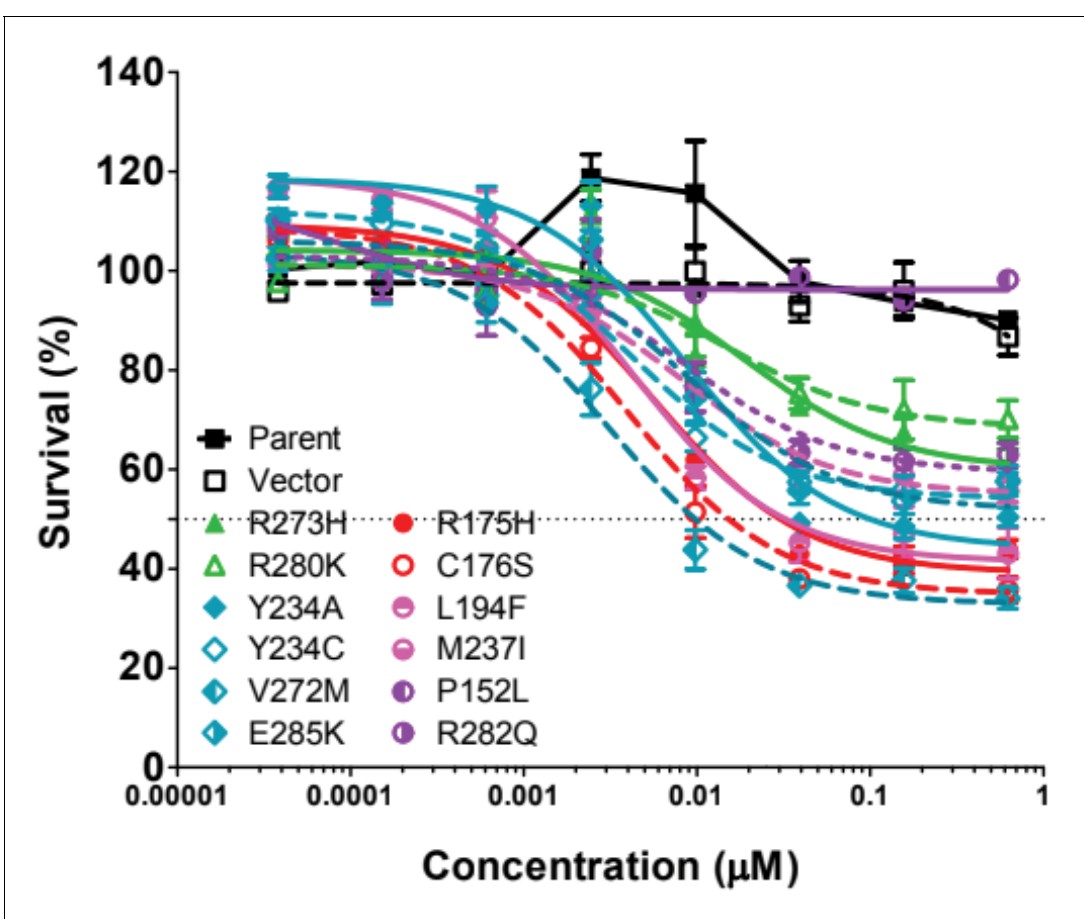

**Figure 5.** Cell growth inhibition of p53 mutants by ZMC1. 12 common p53 mutants were generated by site-directed mutagenesis and expressed in p53-null H1299 cells. Cells were treated with ZMC1 and the cell growth inhibition was measured by Calcein AM assay. $EC_{50}$ values were calculated using nonlinear-fit curves by GraphPad Prism software.

**Table 1.** ZMC1-mediated toxicity and p53 refolding in H1299 cells expressing p53 mutants. $EC_{50}$ data and fits are shown in **Figure 5** and antibody-monitored refolding data are shown in **Figure 6A** and **Figure 7**. [a]Yes, PAB240 staining level was reduced significantly (Fig. after ZMC1 treatment; No, PAB240) staining level was not significantly changed after ZMC1 treatment. Sample size: 3; replicates: 2–3 independent experiments; outliers/exclusions: no.

| Mutant | $EC_{50}$ (nM) | p53 refolding[a] |
|---|---|---|
| Untransfected cells | >1000 | Not applicable |
| Empty vector | >1000 | Not applicable |
| Zinc-binding class | | |
| R175H | 5 | Yes |
| C176S | 2 | Yes |
| L194F | 5 | Yes |
| P152L | >1000 | No |
| R282Q | >1000 | No |
| Stability class | | |
| Y234A | 10 | Yes |
| Y234C | >1000 | No |
| V272M | 2 | Yes |
| E285K | >1000 | No |
| Mixed zinc-binding/stability class | | |
| M237I | >1000 | Yes |
| DNA-binding class | | |
| R273H | >1000 | Yes |
| R280K | >1000 | No |

R282Q and P152L are not detectable in the tested range, suggesting that they are not functionally reactivated by ZMC1. R249S belongs to the zinc-binding class and we previously found that Hs700T cells (p53[R249M]) were not sensitive to ZMC1 (**Yu et al., 2014**). It is possible that mutations of R249, being adjacent to the DNA-contact residue R248, distort the structure of the DNA-binding groove regardless of zinc-binding status. The negative control group consists of the DNA-contact mutants R273H and R280K. As expected, these variants show no increased sensitivity to ZMC1, indicating that their DNA-binding defects cannot be ameliorated in a zinc-dependent fashion.

The most discriminating test of the thermodynamic model is whether the pure stability-class (Y234C, Y234A, V272M, E285K) and mixed stability/zinc-binding class (M237I) of p53 mutants can be reactivated by elevating intracellular zinc. V272M is marked by pronounced ZMC1 sensitivity, with an $EC_{50}$ value (0.002 μM) lower than that observed for any of the zinc-binding class mutants. This result demonstrates that a mutation whose sole consequence is to promote unfolding of p53 can be rescued by zinc. Y234C, E285K, and M237I, however, fail to show ZMC1 sensitivity. The negative result for Y234C agrees with the biophysical data (**Figure 3—figure supplement 2**), which suggested that the extra Cys residue forms a competing, non-native zinc-binding site in the unfolded state. We tested that hypothesis using the Y234A mutant. As predicted by the zinc misligation model, the $EC_{50}$ value of Y234A drops to a value comparable to that of the pure zinc-binding mutants R175H and L194F (**Table 1**). To confirm the result for M237I, we tested two human cancer cell lines that express p53[M237I]. One cell line (T98G) was insensitive to ZMC1 and the other (SUM149PT) showed a partial response in which cell viability remained at ~40% at 10 μM ZMC1 (**Figure 7—figure supplement 1B**). The BXPC3 human cancer cell line bearing p53[Y220C] was also insensitive to ZMC1 (**Figure 7—figure supplement 1B**).

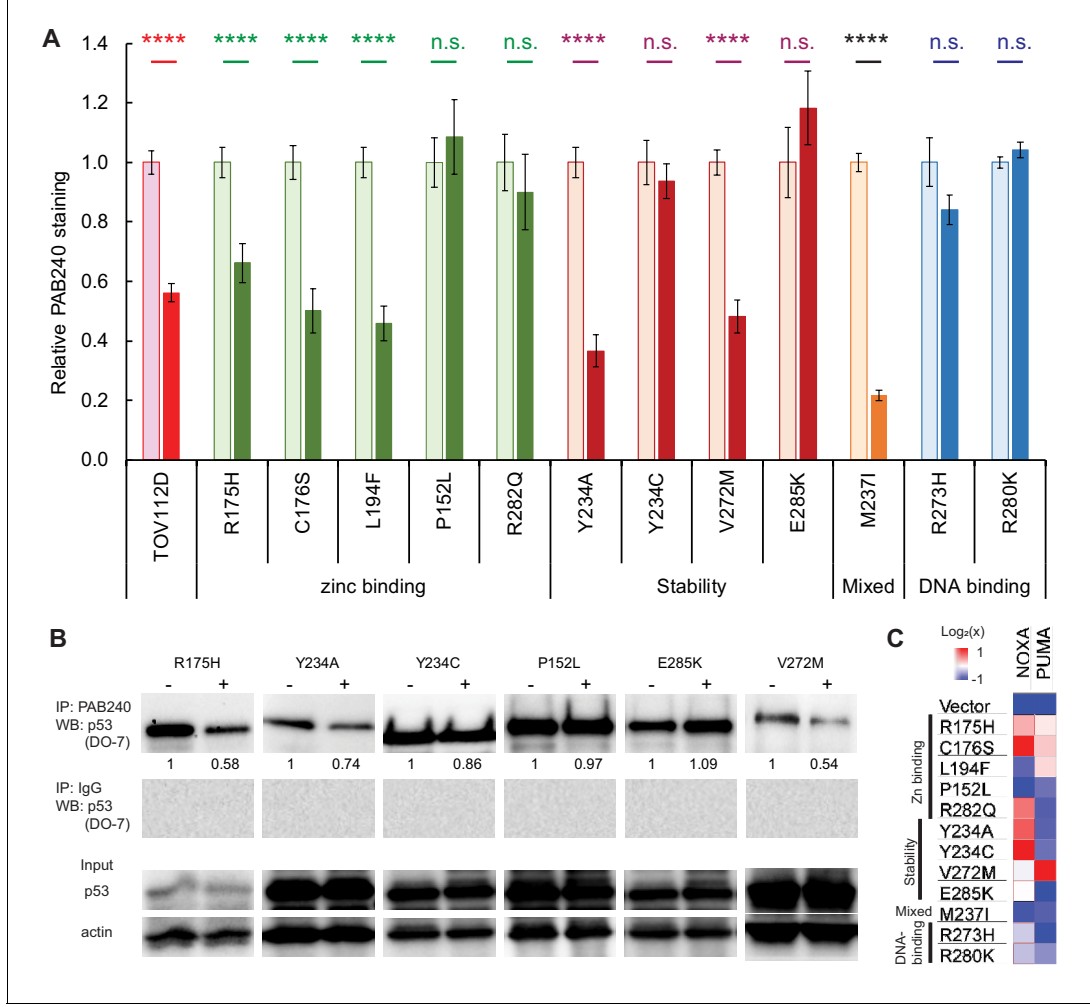

**Figure 6.** Response of p53 mutants to ZMC1 treatment in cells. (**A**) ZMC1-induced folding of p53 mutants quantified by PAB240 immunofluorescence. H1299 cells were stably transfected with p53 mutants and treated with 1 μM ZMC1 (dark bars) or DMSO vehicle (light bars) for 4 hr. TOV112D (p53[R175H]) cancer cell line and parental cell line are positive and negative controls, respectively. ****, p<0.0001; n.s., not significant. Exact p-values are in *Supplementary file 1C*. Sample size: 2; replicates: two independent experiments; outliers/exclusions: no. (**B**) ZMC1-induced folding of p53 mutants quantified by PAB240 IP. Protein lysates were extracted from cells, immunoprecipitated with PAB240, and blotted with the pan-p53 antibody DO-7. Sample size: 1; replicates: two independent experiments; outliers/exclusions: no. (**C**) Activation of *PUMA* and *NOXA* expression by ZMC1 (1 μM, 24 hr) in stably transfected H1299 cells quantified by RT-PCR. Color shades indicate 2-fold differences relative to the vehicle-only controls. Sample size: 3; replicates: two independent experiments; outliers/exclusions: no.

The online version of this article includes the following figure supplement(s) for figure 6:

**Figure supplement 1.** Binding of p53M237I to *CDKN1A* promoter DNA.

**Figure supplement 2.** Folding of p53 protein in V138 cells at 37°C with ZMC1.

## p53 refolding in cells monitored by conformation-specific antibody

To further test the thermodynamic model, and to gain insight as to why some mutants fail to regain their cell-killing functions in the presence of ZMC1, we assayed the extent to which elevated zinc can refold mutant p53 in cells. Refolding was monitored by the reduction in staining by the antibody PAB240, which binds to a cryptic epitope only exposed on p53 unfolding or misfolding (residues 212–217 [*Stephen and Lane, 1992*]). For the zinc-binding class of mutants, the variants that exhibit low $EC_{50}$s (R175H, C176S, L194F) show reduced binding to PAB240 after ZMC1 treatment (*Figure 6A* and *Figure 7*), implying that they undergo zinc-mediated refolding. The human cancer

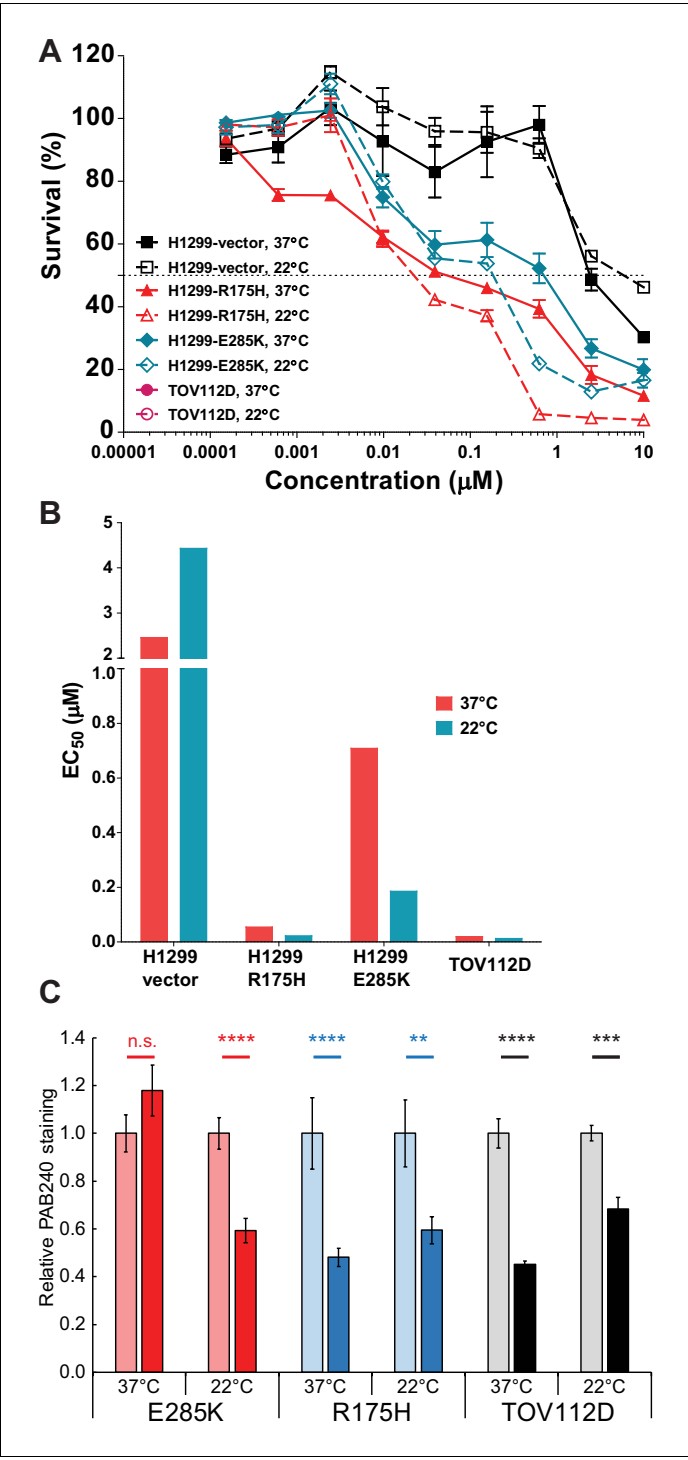

**Figure 7.** p53 folding landscape at lower temperature in cells. (**A**) Decreasing T synergizes with ZMC1 to kill cells expressing mutant p53. Cells were treated with 1 μM ZMC1, incubated at 37°C or 22°C for 4 hr, incubated at 37°C for 72 hr, then assayed for viability by Calcein AM. Values are mean ± SE. Sample size: 3; replicates: 2–3 independent experiments; outliers/exclusions: no. (**B**) EC$_{50}$ values from the curves in panel A. (**C**) Decreasing T synergizes with ZMC1 to refold mutant p53 in cancer cell lines. Experimental protocol is the same as in *Figure 6A*. **, p<0.01; ***, p<0.001; ****, p<0.0001; n.s., not significant. Exact p-values are in *Supplementary file 1D*. Sample size: 2; replicates: two independent experiments; outliers/exclusions: no.

The online version of this article includes the following figure supplement(s) for figure 7:

**Figure supplement 1.** Response of p53 mutant cells to ZMC1.

cell line T47D bearing the p53[L194F] mutation also showed refolding of mutant p53 by reducing PAB240 staining and increasing PAB1620 staining (*Figure 7—figure supplement 1C*). Antibody staining of P152L and R282Q did not change after ZMC1 treatment, consistent with their lack of sensitivity to the drug (*Figure 6A*). We also found that ZMC1 did not refold R249M in Hs700T cells (*Yu et al., 2014*). It is possible that these mutants misfold and/or aggregate in the cell to a conformation that is not amenable to zinc binding or refolding. Another potential explanation for why ZMC1 does not refold certain mutants in cells when there is evidence of refolding in vitro is related to zinc homeostatic mechanisms. Zinc homeostatic genes can be dysregulated in cancer, for example in breast cancer subtypes (luminal, basal, triple-negative) in which cellular zinc levels are elevated compared to normal mammary epithelial cells, and expression of zinc homeostatic genes is perturbed (*Chandler et al., 2016*). Overexpression of metallothioneins or zinc exporting proteins (ZnTs), which muffle ZMC1's metallochaperone activity, may also provide a mechanism of resistance. As expected, staining intensities of the DNA-contact mutants R273H and R280K do not change after ZMC1 treatment. R280K cells appear dark in the absence of ZMC1, consistent with the DNA-contact phenotype of this mutant, but R273H stains brightly. This result suggests that the R273H mutation induces misfolding or aggregation in addition to loss of DNA-binding affinity by direct contact.

For the pure stability-class mutants, the PAB240 refolding results (*Figure 6A*) agree well with the EC$_{50}$ data (*Table 1*): the mutants that regain cell-killing activity in the presence of ZMC1 (Y234A and V272M) show reduced PAB240 antibody staining, and the mutants that are insensitive to ZMC1 (Y234C and E285K in H1299 cells and Y220C in a cancer cell line (*Figure 7—figure supplement 1B*)) react equally well with PAB240 before and after drug treatment. The mixed stability/zinc-binding mutant M237I is an interesting exception. M237I shows the greatest decrease in PAB240 staining after ZMC1 treatment of all mutants tested, yet M237I cells are insensitive to the drug. Increasing intracellular zinc therefore appears to refold M237I, as predicted by our model, but fails to restore its apoptotic activity. In agreement, the human cancer cell lines with p53[M237I] mutation also showed refolding after treatment (by decreasing PAB240 staining after treatment of ZMC1) (*Figure 7—figure supplement 1E*) but was not sensitive to ZMC1 (*Figure 7—figure supplement 1B*). The M237I mutation may compromise p53 function by an additional mechanism such as introducing a structural defect in the folded protein or perturbing a binding interaction between p53 and another protein. We further tested the DNA-binding of p53[M237I] to the p53RE in *CDKN1A* promoter in cells using a luciferase reporter assay and found that the M237I mutant was unable to activate transcription (*Figure 6—figure supplement 1*).

As immunofluorescence is sometimes criticized for its possible interference with protein structure by fixation of cells, we sought to confirm refolding of the p53 mutants by performing immunoprecipitation using PAB240 followed by western blot for p53 proteins. Consistent with the immunofluorescent staining of the individual cells, R175H, Y234A, V272M showed decreased p53 protein by PAB240 pull-down after ZMC1 treatment, while the p53 band intensities remained similar for Y234C, P152L and E285K (*Figure 6B*). As a positive control, we treated the temperature-sensitive A138V with ZMC1 and observed a 2-fold decrease in PAB240 pull-down (*Figure 6—figure supplement 2*).

## Restoration of p53 transcriptional function

To determine if the conformational change observed with the mutants results in restoration of WT p53 transcriptional function, we compared mRNA levels of the p53-responsive genes *PUMA* and *NOXA* before and after 24 hr of ZMC1 treatment, using the transfected cells described above. Mutants for which no increase in transcription is observed for either of the two probe genes (blue or white bars in *Figure 6C*) are P152L and E285K (stability class), M237I (mixed class), and R273H (DNA-contact class). All these mutants are insensitive to ZMC1-mediated cell killing and fail to refold as judged by the PAB240 test (except M237I). Mutants for which ZMC1 treatment enhances transcription of at least one gene (one or more beige or red bars in *Figure 6C*) are R175H, C176S, L194F, R282Q (zinc-binding class), Y234A, Y234C, and V272M (stability class), and R280K (DNA-contact class). Thus, all the mutants that exhibit low EC$_{50}$ values and undergo ZMC1-dependent refolding also have their transcriptional activities at least partially restored by the drug. Similarly, the human cancer cell line with p53[L194F] also showed decreased mutant p53 protein (*Figure 7—figure supplement 1C*) and the induction of the gene expression of *CDKN1A* and *PUMA* (*Figure 7—figure supplement 1E*), indicating reactivation of p53 transcriptional function. The transcriptional assay, however, also finds that ZMC1 elevates mRNA levels of at least one gene for two stability mutants

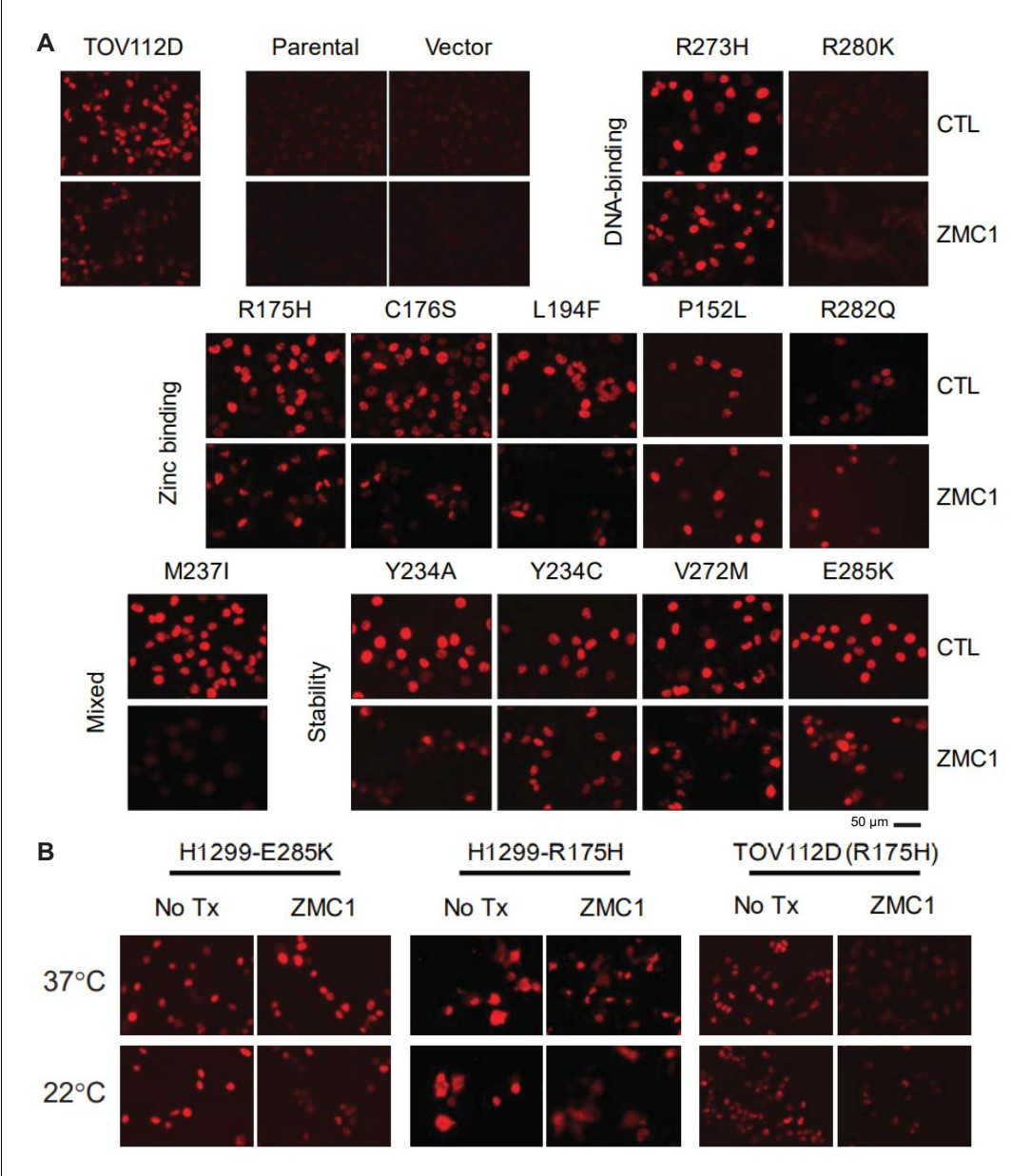

**Figure 8.** Refolding of p53 mutants after ZMC1 treatment. (**A**) H1299 cells were stably transfected with 1 of 12 p53 mutants or an empty vector and treated with ZMC1 for 4 hr. p53 conformation was probed by IF using PAB240, which binds selectively to unfolded p53. TOV112D (p53 R175H) and the parental cell line were used as unfolded and native conformation controls respectively. (**B**) p53 refolding of R175H and E285K with ZMC1 and different incubation temperatures. Sample size: 2; replicates: two independent experiments; outliers/exclusions: no.

(Y234C and R282Q; *Figure 6C*) that failed to show enhanced cell killing or refolding in the presence of ZMC1. The increase in mRNA levels appears to be insufficient to bring about apoptosis.

## V272M and E285K stability mutants

The stability mutant V272M is a particularly interesting case because it is efficiently refolded by ZMC1 (*Figure 6A*) and is threefold more sensitive to the drug than the pure zinc-binding mutant R175H (*Figure 8*). This behavior is consistent with the thermodynamic model. The energy landscape of R175H indicates that raising intracellular $[Zn^{2+}]_{free}$ from 100 pM to 100 nM increases the

percentage of folded R175H from 0.0028% to 2.8% (*Figure 3B*). V272M is destabilized by 2.0 kcal mol$^{-1}$ and if we assume that this relatively small perturbation does not change the enthalpy of unfolding, then V272M will achieve 2.8% refolding at only 10–20 nM [Zn$^{2+}$]$_{free}$. Both IF and IP confirm ZMC1 refolds V272M, whereas ZMC1 fails to induce refolding of the negative control (R273H) (*Figure 6A*; *Figure 6B*). Finally, ZMC1 elicits high level of *PUMA* and *NOXA* expression in V272M (*Figure 6C*). These findings demonstrate that V272M is robustly activated by ZMC1.

E285K, the most severe of the stability class of mutants, failed to refold in the presence of ZMC1 (*Figure 6A*; *Figure 6B*) and cells transfected with *p53*$^{E285K}$ were not sensitive to the drug (*Table 1*). We speculated that the extreme instability of E285K prevented it from refolding despite the elevated intracellular zinc concentrations afforded by ZMC1 treatment, as well as its increased affinity for most p53REs relative to WT. We asked whether reducing temperature could act synergistically with ZMC1 to reactivate E285K. The temperature of p53$^{E285K}$ H1299 cell cultures was lowered to 22° C for 4 hr in the presence of ZMC1 to allow for temperature-assisted refolding, then returned to 37° C for cell-killing assays. This procedure resulted in 3.8-fold decrease in EC$_{50}$ compared to the control in which 37°C was maintained throughout (*Figure 7A*, *Figure 7B*). Of note, we observed a similar effect for R175H in H1299 (2.4-fold decrease) and TOV112D cells (1.6-fold decrease), suggesting that low temperature acts synergistically with zinc-binding and stability mutants alike, as predicted by the thermodynamic model. We further determined that reducing temperature to 22°C successfully induced ZMC1-mediated refolding of E285K (*Figure 7C*).

## Restoration of p53 function in vivo

ZMCs are currently in pre-clinical development and represent a viable strategy to reactivate mutant p53 in the clinic. One of the attractive features of the ZMC program is that the spectrum of patients that will potentially respond to the drugs is known (those that harbor zinc-binding class p53 mutations). To determine if this spectrum should include individuals that have stability-class mutations, we sought to obtain pre-clinical evidence for this using the xenograft tumor assay. We treated the animals with the tumors derived from the stable cell lines with V272M and E285K mutations with ZMC1 and found that the growth of the V272M tumors was inhibited but growth of E285K tumors was not (*Figure 9*; *Figure 9—figure supplement 1*), implying that ZMC1 can be used to reactivate certain p53 stability mutations, in addition to the zinc-binding deficient mutations.

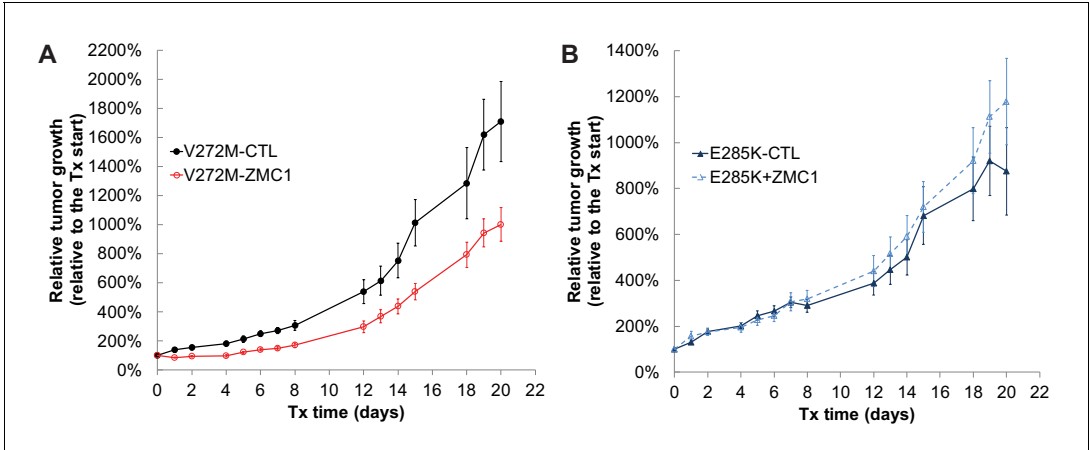

**Figure 9.** In vivo efficacy of ZMC1 in stability-class mutants V272M (**A**) and E285K (**B**). Mice bearing human xenograft tumors were treated with ZMC1 (5 mg/pk daily, IP) or DMSO vehicle. See Methods for treatment and allocation details. Treatment and control groups were n = 14 and n = 12, respectively, for both V272M and E285K xenografts. All sites exhibited tumor growth and none were excluded from analysis.

The online version of this article includes the following figure supplement(s) for figure 9:

**Figure supplement 1.** Individual tumor growth curves for the processed data presented in *Figure 5*.

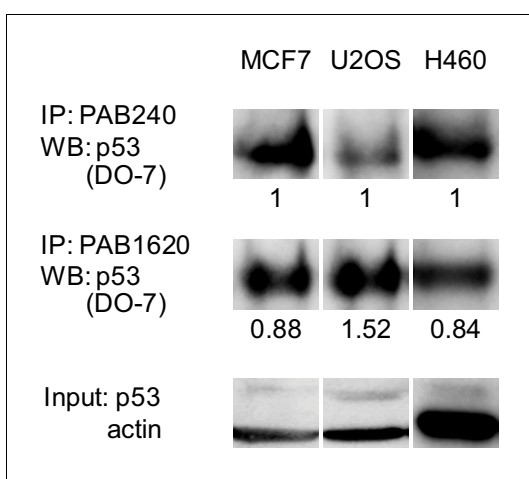

**Figure 10.** A significant fraction of WT p53 is unfolded in cells. WT p53 cell lines MCF7, U2OS, and H460 were lysed and immunoprecipitated using PAB240 or PAB1620 antibodies. Sample size: 1; replicates: two independent experiments; outliers/exclusions: no.

## WT p53 is balanced between folded and unfolded states

We hypothesized from the energy landscape map that at physiological conditions of temperature and available zinc concentration, WT p53 may exist in approximately equal populations of folded and unfolded molecules. To address this question in context of the cell, we employed three cell lines expressing WT p53: MCF7, U2OS, and H460. We immunoprecipitated p53 protein from cell lysates with PAB240 or PAB1620 and blotted with the pan-p53 antibody DO-7 (*Figure 10*). Because the two antibodies may have different affinities for p53, the percentages of unfolded and folded p53 cannot be determined quantitatively from this experiment. The intensities of the PAB240 bands, however, are comparable to those of the PAB1620 bands, suggesting that a significant fraction of WT p53 is unfolded in all three cell lines.

## Discussion

Our thermodynamic model of p53 folding is derived from two measurable properties: the free energy of DBD folding in the absence of metal, and the binding affinity of zinc to the folded protein. Of the 22 most common tumorigenic mutations examined here, all but three reduce stability by >1 kcal mol$^{-1}$, decrease $Zn^{2+}$ binding affinity by >10 fold, or both. The remaining three belong to the DNA-contact class. These findings suggest that loss of thermodynamic stability and/or metal-binding affinity play a dominant role in p53-related cancers, and underscore p53's remarkable sensitivity to missense mutation at nearly every codon position.

In cells, ZMC1 reactivated three of the zinc-binding mutants (R175H, C176S, L194F) for cell killing (*Table 1*), PAB240-monitored refolding (*Figure 6A*), and *PUMA/NOXA* transcription (*Figure 6C*). We previously demonstrated that the zinc-coordinating mutants (C238S and C242S) were also reactivated by ZMC1, while the R249M mutant was not, due to a lack of wild-type conformation induction (*Yu et al., 2014*). Taken together, five of the eight zinc-binding class mutants were reactivated by ZMC1 in cells. The biophysical data are thus correlated with the biological results but there are some discrepancies. The energetic modeling does not consider structural effects, so it is expected that some mutants will remain inactive in the cell even if proper stability and zinc binding are restored by metallochaperones. For example, the R282Q mutation, being in the same helix as the DNA-contact mutation R280K, is likely to perturb the structure of the DNA-binding pocket. ZMC1 also failed to functionally rescue the mixed zinc-binding/stability mutant M237I, causing a WT conformation change (*Figure 6A*) but not an induction of *PUMA/NOXA* transcription (*Figure 6C*). One possible explanation is that certain mutations may cause p53 to be sequestered in the cytoplasm and ZMC1 may not affect this mislocalization.

One of the primary predictions of the model is that increasing zinc concentration will refold stability class as well as zinc-binding class mutants. The archetypical example of small-molecule-induced p53 stabilization is binding of PhiKan and similar compounds to the surface cavity on Y220C left by the Tyr220→Cys alteration (*Boeckler et al., 2008*; *Liu et al., 2013*; *Baud et al., 2018*; *Bauer et al., 2019*). These drugs offer the potential of bio-orthogonality but bind to only a single p53 mutant and do so with (thus far) weak affinity ($K_d$ = 1–100 μM). Zinc-induced stabilization, by contrast, takes advantage of an existing high-affinity site on p53 ($K_{Zn} \sim 10^{-15}$ M) and can in principle refold any stability-class mutant, but suffers from lack of bio-orthogonality. As evidence for the potential breadth of ZMC therapy, treating cells with ZMC1 reactivated three stability-class mutants in cells (Y234A, V272M, and G245S [*Yu et al., 2014*]) and in vivo (V272M), failed to reactivate two (Y234C and E285K). Y234C, like all the Tyr→Cys mutants that we tested, is refractory to zinc-induced stabilization most likely because of metal misligation in unfolded or partially folded states. E285K is the most unstable variant that we have characterized, which may explain why elevated zinc alone was insufficient for refolding without the additional stabilizing factor of reduced temperature. Nonetheless, the discovery that ZMC's can reactivate mutants beyond just the class of zinc deficient mutation is a significant finding.

Previous work by our laboratory as well as that of Fersht established that the stability of DBD is low at 37°C. The new insight offered by the current study is that DBD achieves this instability by a unique mechanism. In the absence of zinc, we find that WT apoDBD is much more unfolded than previously thought at body temperature ($\Delta G_{apo}$ = 6.9 kcal mol$^{-1}$). This is not due to DBD being intrinsically disordered—it is quite stable at 10°C—but to the anomalously high dependence of $\Delta G_{apo}$ on temperature. Offsetting DBD's inherent propensity to unfold is its extraordinary affinity for zinc ($K_{Zn}$ = 1.6×10$^{-15}$ M), one of the highest yet reported for any eukaryotic protein. These biophysical data suggest that, in the absence of other cellular considerations such as chaperones and p53 binding partners, a dominant factor determining whether p53 is folded or unfolded is the available concentration of cytosolic zinc At typical intracellular concentrations of available Zn$^{2+}$ (10$^{-10}$ M), our modeling predicts that the folded and unfolded populations are comparable—a prediction supported by conformation-specific antibody experiments (*Figure 10*). This balance may explain why so many different missense mutations at nearly every codon position in the DBD gene are associated with loss of p53 function and cancer (*Baugh et al., 2018*). More often than not, an amino acid substitution at any given position will decrease folding free energy rather than increase it, and loss of thermodynamic stability is the major cause of diseases that are caused by missense mutation of a single protein (*Yue et al., 2005*). To emphasize this point, nearly all mutants examined in this study destabilize DBD and/or decrease its zinc affinity. The remainder are DNA-contact mutants, which can be reliably deduced from inspecting the X-ray crystal structure of DBD.

Why, then, might p53 have evolved with this unusual and precarious combination of high instability and zinc affinity? One explanation is that it constitutes a built-in failsafe to help rein in p53's powerful cytotoxic activities should cellular checkpoint pathways become compromised. Carrying this scenario further, one might also speculate that it is a mechanism by which the cell can reversibly regulate p53 function by modulating its conformation through available zinc levels. Evidence for conformational regulation of p53 has existed for some time. Milner and Watson reported that fresh medium induced cell cycle and conformational changes in p53 (*Milner and Watson, 1990*). Hainaut, Milner, and colleagues reported that incubating cells and cell lysates with metal chelators can starve p53 of Zn$^{2+}$ and induce the PAB240-binding form of p53, which can be rescued by re-introducing zinc (*Méplan et al., 2000*; *Hainaut and Milner, 1993*). They also reported that oxidative agents when applied to cells can also induce the PAB240 conformation in WT p53. Oxidative agents increase intracellular free zinc levels by oxidizing the zinc-binding cysteine residues on cytosolic metallothionein proteins, decreasing their affinity for zinc (*Maret, 2017*). Since stress signals can regulate p53 it is also possible that similar stress signals could modulate the relative abundance of holo and apo p53 in the cell by altering zinc affinity or intracellular zinc levels.

Approximately 10% of the proteins encoded by the human genome bind zinc; however, not until the last decade have researchers discovered that cytosolic zinc levels are in the picomolar range while the total cellular zinc is in the hundreds of micromolar (*Krezel and Maret, 2006*) and more importantly, that many proteins are functionally regulated by zinc (*Maret, 2017*). This explains why there exists a complex repertoire of cellular homestatic genes consisting of cellular importers (ZIPs), antiporters (ZnTs) and cytosolic zinc buffers (metallothioneins). Proteins such tyrosine phosphatases

are regulated by zinc, and these proteins typically have zinc affinities in the range of cytosolic free zinc (*Wilson et al., 2012*). Although $K_{Zn}$ of apoDBD is extremely low ($1.6 \times 10^{-15}$ M), the effective $K_{Zn}$ at physiological temperature ($K_{Zn,eff}$) is orders of magnitude higher owing to the inherent instability of apoDBD at 37℃. $K_{Zn,eff}$ is approximated by the product of $K_{Zn}$ and $K_{apo}$ (*Equation 4*)—a value that our modeling suggests is close to cytosolic $[Zn^{2+}]_{free}$ (*Figure 3A*). This suggests that physiologic perturbations in cytosolic zinc levels (100's of pM), could in theory modulate zinc binding to p53 and hence its function. Moreover, given our findings that both apo and holo forms of WT p53 can be detected in cells, this suggests that p53 could potentially be regulated conformationally by zinc.

In conclusion, we have quantified the folding free energies and zinc-binding affinities of the 22 most prevalent p53 mutations in cancer, many of which have not previously been characterized. Our thermodynamic modeling places the mutations into three distinct classes that will be useful to stratify patients for potential zinc metallochaperone treatment. We have demonstrated that ZMC1 treatment rescues the function of not only zinc-binding class mutants (e.g. R175, C176, H179, C242), but also that of some stability-class mutants (e.g. G245 and V272). Mutations in these six positions alone are associated with new cancer cases in more than 120,000 patients each year in the U.S. (*Siegel et al., 2020*).

# Materials and methods

**Key resources table**

| Reagent type (species) or resource | Designation | Source or reference | Identifiers | Additional information |
|---|---|---|---|---|
| Antibody | Anti-p53 PAB240 (mouse monoclonal) | EMD Chemicals | OP29 | 1:400 for IF, 2 µg for IP |
| Antibody | Anti-p53 PAB1620 (mouse monoclonal) | EMD Chemicals | OP33 | 1:50 for IF, 2 µg for IP |
| Antibody | Goat anti-mouse (goat polyclonal, HRP conjugate) | Santa Cruz Biotechnology | sc-2005 | 1:3000 for western blot |
| Antibody | Anti-p53 (DO-1) (mouse monoclonal) | Santa Cruz Biotechnology | sc-126 | 1:1000 for western blot |
| Antibody | Anti-p53 (DO-7) (mouse monoclonal) | Santa Cruz Biotechnology | sc-47698 | 1:1000 for western blot |
| Antibody | Anti-beta-actin (AC-15) (mouse monoclonal) | Santa Cruz Biotechnology | sc-69879 | 1:2000 for western blot |
| Commercial assay or kit | Q5 Site-Directed Mutagenesis Kit | NEB | E0554S | Site-directed mutagenesis |
| Sequence-based reagent | TaqMan assay human p21 | ThermoFisher | Hs00355782_m1 | |
| Sequence-based reagent | TaqMan assay human PUMA | ThermoFisher | Hs00248075_m1 | |
| Sequence-based reagent | TaqMan assay human NOXA | ThermoFisher | Hs00560402_m1 | |
| Sequence-based reagent | TaqMan assay human beta-actin | ThermoFisher | Hs99999903_m1 | |
| Chemical compound, drug | ZMC1 | Synthesized in this work | | |
| Genetic reagent (*M. musculus*) | Female CR ATH HO (order when 6–8 weeks old) | Charles River Laboratories | | |

## Samples size and data analysis

Samples sizes were adjusted to provide smooth interpolation across the experimental range; experimental range spanned at least one order of magnitude greater the highest and lower than the smallest central value (e.g. $K_d$, $C_m$, $EC_{50}$) unless constrained by physical limitations (solubility, buffer range, etc.). Number and type of experimental replicates are indicated in figure legends. Cell biological replicates consist of separate plates of cells from a common source. Data from replicate experiments were pooled prior to analysis. All data are shown and excluded outliers are marked. In nonlinear data, potential outliers were identified on inspection, and excluded if they met both of the following conditions: (1) removal of the outlier improved goodness of fit as judged by the average relative standard error of the fit parameters, and (2) upon exclusion and re-fitting, the excluded point lies outside the 95% prediction interval of the fit. For multiple outliers, the Holm-Bonferroni correction was applied when calculating the 95% prediction interval. R code for outlier testing is available at https://codeberg.org/AlanBlayney/mutant-p53. Curve fitting and data plotting were performed with SigmaPlot 13.0, GraphPad Prism, KaleidaGraph 4.5, and R 3.6.1, with additional annotation in Adobe Illustrator. Three-dimensional plots were generated in R using the 'lattice' package (*R Development Core Team, 2020*; *Sarkar, 2008*).

## Protein purification and preparation of zinc solutions

DBD genes (p53 residues 94–312, no expression or purification tags) were cloned into pET23a plasmids and transformed into *E. coli* BL21(DE3) cells. Cultures were grown at 37°C to $OD_{600}$ = 0.6, induced with isopropyl β-D-1-thiogalactopyranoside, and expressed for 12–15 hr at 18°C. Following cell lysis and centrifugation, soluble DBDs were purified in 20 mM Tris (pH 7.2), 5 mM β-mercaptoethanol using SP-sepharose then heparin HiTrap cation exchange columns (GE Life Sciences) (11, 21). Mutant proteins were judged to be > 98% pure by SDS-PAGE with Coomassie brilliant blue staining. Apo proteins were generated by lowering pH to 4.8 using acetic acid in the presence of a large excess of EDTA, restoring pH to 7.5 using Tris, and removing EDTA and zinc with a DG10 desalting column (Bio-Rad) (9). Full-length p53 was expressed as a fusion construct with a cleavable N-terminal expression tag consisting of HisTag fused to ribose-binding protein derived from *Thermoanaerobacter tengcongensis*. FL-p53 was purified by nickel-NTA chromatography (Qiagen) following manufacturer's protocols. The HisTag and ribose-binding protein purification tag was then removed using human rhinovirus 3C protease, at which point FL-p53 was purified identically to DBD with the addition of a final clean-up step on a Superdex S200 size exclusion column (GE Life Sciences). FL-p53 w*as* >90% pure by SDS-PAGE with Coomassie brilliant blue staining. All in vitro experiments were performed in 50 mM Tris (pH 7.2), 0.1 M NaCl, 10 mM β-mercaptoethanol unless otherwise noted.

$ZnCl_2$ stocks were dissolved in 30 mM HCl and their concentrations determined by titration with 4-(2-pyridylazo)resorcinol using $\varepsilon_{500}$ = 66,000 $M^{-1}$ $cm^{-1}$ for the PAR-$Zn^{2+}$ complex. $[Zn^{2+}]_{free}$ concentrations were fixed at the indicated values by mixing 2 mM chelator (EDTA, N-(2-Hydroxyethyl) EDTA, EGTA, or diethylenetriaminepentaacetic acid) with 0.005–1.75 mM $ZnCl_2$. All chelators were purchased from Sigma Aldrich (St. Louis, MO). These total zinc concentrations were always in excess of protein concentration (1 μM). $[Zn^{2+}]_{free}$ values were calculated using the MAXCHELATOR software suite (http://somapp.ucdmc.ucdavis.edu/ pharmacology/bers/maxchelator/). Concentrations of $Zn^{2+}$-chelators were determined by titrating the chelator solutions against pre-formed PAR-$Zn^{2+}$ complex and measuring the decrease in absorbance in buffer. Concentrations of FZ3 solutions (Life Technologies, Norwalk, CT) were determined by equivalence point titration with $Zn^{2+}$ in buffer.

## Protein stability, zinc-binding, and DNA-binding assays

Samples for urea denaturation experiments were prepared using a Hamilton Microlab 500 diluter, with final urea concentrations measured by index of refraction (*Butler and Loh, 2003*). For urea denaturation studies in the presence of zinc, samples contained chelator and $ZnCl_2$ at the concentrations indicated above. Trp fluorescence was measured using either a Fluoromax-4 spectrofluorometer (Horiba Scientific, Edison, NJ) in a 5 mm x 5 mm quartz cuvette or in a SpectraMax i3x plate reader (Molecular Devices, Sunnyvale, CA) in 96-well UV-Microplates (Thermo Scientific, Waltham, MA) with comparable results ($\lambda_{ex}$ = 280 nm, $\lambda_{em}$ = 355 nm). The resultant curves were fit to the 2-state linear extrapolation model with linear baselines to obtain ΔG and *m*-values (*Pace, 1975*; *Schellman, 1975*). To increase the accuracy of our measurements, *m*-values of all curves were pooled

($m_{pool}$ = 3.10 ± 0.12 kcal mol$^{-1}$ M$^{-1}$, n = 213, mean ± SE) and ΔG was calculated from the midpoint of denaturation ($C_m$) of each curve, which is more accurately determined than $m$, according to the equation ΔG = $C_m$·$m_{pool}$. ΔG values were then fit to *Equation 3* to obtain ΔG$_{apo}$ and $K_{Zn}$. For variable temperature urea denaturation studies, samples were equilibrated at the indicated temperature for 15–24 hr, either in urea or GdnHCl denaturant. ΔG values were obtained by the pooled *m*-value method described above (in GdnHCl, $m_{pool}$ = 7.01 ± 0.92 kcal mol$^{-1}$ M$^{-1}$, n = 6, mean ± SE). ΔG values were plotted as a function of T and fit to the Gibbs-Helmholtz equation (*Figure 2—figure supplement 1C*). $K_{Zn}$ competition assays were carried out by incubating unfolded apoDBD (6 M urea) at the indicated concentrations with 15 nM ZnCl$_2$ and 30 nM FZ3 for 1 hr at room temperature. FZ3 fluorescence was then scanned and IC50 was obtained by fitting the data to *Equation 6*:

$$F_{obs} = \frac{A}{1 + \exp\left(\frac{IC50 - x}{b}\right)} \tag{6}$$

where $F_{obs}$ is observed fluorescence, $A$ is the curve amplitude, $x$ is log[DBD], and $b$ is an empirical steepness parameter. $K_{Zn}$ was then calculated from the Munson-Robdard solution to the Cheng-Prusoff equation (*Equation 7*; *Munson and Rodbard, 1988*):

$$K_{Zn} = \frac{IC50}{1 + \frac{FZ3(y0+2)}{2K(y0+1)} + y0} - K\frac{y0}{y0+2} \tag{7}$$

where *FZ3* is the total concentration of FZ3, $K$ is the dissociation constant of the FZ3·Zn$^{2+}$ complex (15 nM per the manufacturer), and $y0$ is the ratio of bound FZ3 to free FZ3 in the absence of DBD (equal to unity in our conditions).

$K_{Zn}$ measurements by intrinsic fluorescence were performed by incubating apoDBD or apo-FL-p53 with chelator/ZnCl$_2$ mixtures (as described above) for 16 hr at 10°C. Zinc binding was monitored by the slight increase in fluorescence at 306 nm (DBD) or 350 nm (FL-p53), with excitation at 280 nm. Data for DBD were fit to the single-site binding equation (*Equation 8*):

$$F_{obs} = F_0 + \frac{A[Zn]_{free}}{K_{Zn} + [Zn]_{free}} \tag{8}$$

where $F_0$ is the fluorescence baseline and A is the amplitude of the curve. To determine if there was cooperativity of zinc binding to the FL-p53 tetramer, those data were fit to the Hill-binding equation (*Equation 9*):

$$F_{obs} = F_0 + \frac{A\left([Zn]_{free}\right)n}{(K_{Zn})n + \left([Zn]_{free}\right)n} \tag{9}$$

where n is the Hill parameter. Reported $K_{Zn}$ values and Hill parameters were averaged from independent replicates.

$K_{DNA}$ values were obtained using 5'-Cy3 labeled oligonucleotides (Eurofins Genomics, Louisville, KY; see *Supplementary file 1A* for sequences). Annealed oligonucleotides (50 nM) were incubated with 5–10,000 nM DBD on ice for 1 hr in 50 mM Tris pH 7.2, 100 mM NaCl, 1 mM TCEP, and 0.005% Tween-20 in black 96-well plates. Uncalibrated fluorescence anisotropy was measured using Spectra-Max i3x equipped with the rhodamine fluorescence polarization module (G-factor = 1, λ$_{ex}$\λ$_{em}$ = 535 nm/595 nm) (Molecular Devices, Sunnyvale, CA). Data for all trials of all mutants for a given sequence were subjected to global curve fitting with *Equation 10*, linking A and n across all trials.

## Energy landscape plots

The folding landscape of WT DBD was calculated from the four-state model, assuming that unfolded DBD binds only a single zinc ion, using *Equation 10*:

$$F_{holo} = \frac{[Zn]_{free}\cdot(K_{Zn})^{-1}}{1 + [Zn]_{free}\cdot(K_{Zn})^{-1} + \frac{1+[Zn]_{free}\cdot(K_{Zn,U})^{-1}}{K_{apo}}} \tag{10}$$

Energy landscapes of the zinc-binding class of DBD mutants were generated from the WT landscape by subtracting the difference in the $\Delta G_{apo}$ value of the mutant from that of WT DBD (both measured at 10°C) at each temperature, and using the mutant $K_{Zn}$ value in *Equation 10*. $K_{Zn}$ values were assumed to be independent of temperature, and $K_{Zn,U}$ values for WT and mutant DBD were assumed to be identical. R code for the generation of landscape plots is available at https://code-berg.org/AlanBlayney/mutant-p53.

## Cell lines, culture conditions, expression vectors, and chemicals

TOV112D, H1299, T47D, SUM149PT, T98G, Calu-3, U2OS and V138 cell lines were cultured in DMEM with 10% FBS. BXPC3, MCF7 and H460 were cultured in RPMI with 10% FBS. TOV112D, H1299, T47D, SUM149PT, T98G, Calu-3, MCF7, U2OS and H460 were purchased from American Type Culture Collection (ATCC). V138A was a gift from Dr. Arnold Levine. Cell lines were authenticated by examination of morphology, genotyping by PCR and growth characteristics. All cell lines were determined to be Mycoplasma free using the Universal Mycoplasma Detection Kit (ATCC 30–1012K, identifier 30–1012K). Mycoplasma testing methods and results are included as *Supplementary file 2*. Vectors for expressing p53 mutants were generated by site-directed mutagenesis using the Q5 Site-directed mutagenesis kit (New England Biolabs), following the manufacturer's instructions. The initial plasmid with human *TP53* R273H mutant was purchased from OriGene. The oligonucleotides for the mutations are listed in *Supplementary file 1B*. ZMC1 was synthesized by Rutgers Molecular Design and Synthesis group, Office of Research and Economic Development (*Zaman et al., 2019*).

## Transfection of plasmid constructs and generation of stable cell lines

Cells at 80–90% confluence were transfected with Lipofectamine 3000 (Thermo Fisher), following the manufacturer's instructions. Expression of p53 protein was confirmed by Western blot. For generation of stable cell lines, H1299 cells were transfected with a vector encoding various p53 mutants generated by site-directed mutagenesis. Cells were then selected for G418 resistance. Single positive clones were isolated and stably maintained in G418-containing medium.

### Cell growth inhibition assays

5,000 cells per well were cultured in 96-well plates such that 50% confluence was reached after one day. At this point, serial dilutions of ZMC1 were added and incubation continued for 3 days. Viability was then measured by Calcein AM (Trevigen, Gaithersburg, MD). Variable temperature incubations were performed as described in the text. Growth inhibition was quantified as absolute $EC_{50}$ by fitting the average data from three replicates to a three-parameter sigmoid.

### Immunofluorescent staining and immunoprecipitation

Immunofluorescent staining was performed as described previously (*Yu et al., 2012*). The conformation of folded and misfolded/unfolded p53 were recognized by the antibodies PAB1620 (1:50) and PAB240 (1:400) (EMD Millipore) respectively, with overnight binding. The secondary antibody, goat anti-mouse IgG, was incubated for 40 m. Fluorescent staining intensity was quantified using ImageJ software (NIH).

For immunoprecipitation, cells were harvested and lysed using RIPA buffer. The lysates were incubated with protein A/G beads with PAB240/PAB1620 or IgG overnight at 4°C. The IP products were detected by Western blot using the pan-53 antibody DO-7. IgG was used as a negative control. The input total lysates were detected by western blot with p53 antibody DO-1 and actin served as the internal loading control. IgG, DO-7, DO-1, actin antibody, and protein A/G beads were purchased from Santa Cruz Biotechnology.

### Luciferase reporter assay

The p53 recognition element in the *CDKN1A* promoter region, constructed in pGL3 vector, was a gift of Dr. Carol Prives (Columbia University, New York, NY). It was transfected into the cells in 96-well plate, followed by the treatment of 1 μM ZMC1 for 48 hr. The luciferase reporter assay was then performed using Dual-Glo luciferase assay system (Promega) following the manufacturer's instructions.

## Mouse experiments

Mice were housed and treated according to guidelines established by the Institutional Animal Care and Use Committee of Rutgers University, who also approved all mouse experiments (animal protocol PROTO99900044, approval date 10/16/2019 – 10/15/2022). Nude mice (NCR nu/nu) were purchased from Taconic Biosciences. Xenograft tumors were generated from the stable tumor cell lines H1299-V272M and H1299-E285K ($1 \times 10^7$ cells/tumor site/mouse). Tumor dimensions were measured every 1–4 d and their volumes (V) were calculated by using the formula: V = (length $\times$ width$^2$ $\times$ $\pi$)/6. Tumors were allowed to grow to 50 mm$^3$ at which point the mice were randomly allocated to treatment and control groups and ZMC1 or vehicle was administered.

## Acknowledgements

We thank Dr. Arnold Levine for the p53$^{A138V}$ cell line and Dr. Carol Prives for the luciferase reporter construct p21short. This work was supported by NIH grants R01 CA200800 and K08 CA172676 (to DRC) and F30 GM113299 (to ARB) and from the Breast Cancer Research Foundation (to DRC).

## Additional information

### Funding

| Funder | Grant reference number | Author |
|---|---|---|
| National Institutes of Health | F30 GM113299 | Adam R Blanden<br>Stewart N Loh |
| National Institutes of Health | R01 CA200800 | Darren R Carpizo |
| National Institutes of Health | K08 CA172676 | Darren R Carpizo |
| Breast Cancer Research Foundation | | Darren R Carpizo |

The funders had no role in study design, data collection and interpretation, or the decision to submit the work for publication.

### Author contributions

Adam R Blanden, Conceptualization, Formal analysis, Funding acquisition, Investigation, Methodology, Writing - original draft; Xin Yu, Conceptualization, Formal analysis, Investigation, Methodology, Writing - original draft, Writing - review and editing; Alan J Blayney, Formal analysis, Investigation, Writing - review and editing; Christopher Demas, Yue Liu, Tracy Withers, Investigation; Jeung-Hoi Ha, Resources, Investigation; Darren R Carpizo, Conceptualization, Formal analysis, Supervision, Writing - original draft, Project administration, Writing - review and editing; Stewart N Loh, Conceptualization, Formal analysis, Investigation, Methodology, Writing - original draft, Project administration, Writing - review and editing

### Author ORCIDs

Alan J Blayney https://orcid.org/0000-0003-0741-1550
Stewart N Loh https://orcid.org/0000-0003-4387-9644

### Ethics

Animal experimentation: Mice were housed and treated according to guidelines established by the Institutional Animal Care and Use Committee of Rutgers University, who also approved all mouse experiments. (animal protocol PROTO99900044, approval date 10/16/2019 - 10/15/2022).

### Decision letter and Author response

Decision letter https://doi.org/10.7554/eLife.61487.sa1
Author response https://doi.org/10.7554/eLife.61487.sa2

# Additional files

## Supplementary files

• Supplementary file 1. Tables of DNA oligonucleotide sequences and exact p-values for p53 refolding in cells. (**A**) Table of DNA sequences used in p53-p53RE binding experiments. (**B**) Table of oligonucleotides used to generate p53 mutants by site-directed mutagenesis. (**C**) Table of exact Student's t-test p-values for p53 refolding in cells, measured by immunofluorescence (*Figure 6A*). (**D**) Table of exact Student's t-test p-values for p53 refolding in cells, measured by immunofluorescence (*Figure 7C*)

• Supplementary file 2. Mycoplasma testing and results.

• Transparent reporting form

## Data availability

All data generated or analyzed during this study are included in the manuscript and supporting files.

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
