## [Decision Letter]

**Acceptance summary:**

The DNA-binding domain (DBD) of p53 is the site of an unusually large number of clinically identified missense mutations that compromise p53's activity, leading to different cancers. The DBD is also unusual in being mostly unfolded (in the absence of Zinc) at body temperature due to its intrinsic marginal stability. Blanden et al. dissect the contributions of mutations and Zinc chaperones to DBD stability, Zinc binding affinity or both, revealing a complex though convincing interplay between these three determinants of DBD activity. They find that p53's Zinc affinity is remarkably high, whereas its body-temperature folding free energy is unfavourable, leading to a nearly perfect cancelling out of the two contributions at physiological conditions. They also examine the ability of Zinc chaperones to promote refolding of commonly observed p53 mutants. This study provides a strong biophysical and biochemical basis that could potentially be useful for personalized oncological regimens for cancer patients who harbor these mutant p53s as ZMC1 is currently at the stage of clinical trials.

**Decision letter after peer review:**

Thank you for submitting your article "Zinc shapes the folding landscape of p53 and establishes a pathway for reactivating structurally diverse cancer mutants" for consideration by *eLife*. Your article has been reviewed by two peer reviewers, including Sarel Jacob Fleishman as the Reviewing Editor and Reviewer #1, and the evaluation has been overseen by Cynthia Wolberger as the Senior Editor. The following individual involved in review of your submission has agreed to reveal their identity: Hua Lu (Reviewer #2).

The reviewers have discussed the reviews with one another and the Reviewing Editor has drafted this decision to help you prepare a revised submission.

Summary:

The DNA-binding domain (DBD) of p53 is the site of an unusually large number of clinically identified missense mutations that compromise p53's activity, leading to different cancers. The DBD is also unusual in being mostly unfolded (in the absence of Zinc) at body temperature due to its intrinsic marginal stability. Previous reports, including by the current authors, provided evidence that DBD stability is impacted by Zinc. Now, Blanden et al. dissect the contributions of mutations and Zinc chaperones to DBD stability, Zinc binding affinity or both, revealing a complex though convincing interplay between these three determinants of DBD activity. They find that p53's Zinc affinity is remarkably high, whereas its body-temperature folding free energy is unfavourable, leading to a nearly perfect cancelling out of the two contributions at physiological conditions. They also examine the ability of Zinc chaperones to promote refolding of commonly observed p53 mutants. This study provides a strong biophysical and biochemical basis that could potentially be useful for personalized oncological regimens for cancer patients who harbor these mutant p53s as ZMC1 is currently at the stage of clinical trials.

Essential revisions:

1) DBD's marginal stability poses special challenges to any molecular study. The authors must make inferences of the DBD's behavior at 37 ^o^C, while the protein cannot be purified in an unaggregated form at this temperature. Their results are therefore based on measurements taken at 2-22 ^o^C, as noted in the subsection “Extrapolation to physiological conditions”. Even if this is standard practice in the p53 field, I think it's important to note whether this risks any of the conclusions drawn by the authors and provide some reasonable bounds on the errors that this assumption may cause. It seems to me that the very interesting (and major) conclusion of this study that the dGfold of p53 at physiological concentrations may be sensitive to small changes in the concentration of Zinc stands on this analysis. It would therefore be good to assess how safe this conclusion is.

2) In the Materials and methods, the authors note that "outliers were identified on inspection and excluded based on gross divergence from the modeled fit". It would be preferrable to see all of the measurements that were done accurately and see the mean and standard errors that result from all of the data. An alternative is for the authors to provide a detailed explanation on the criteria used to exclude data in each figure. I also strongly recommend that any details on the statistical analysis (what type of repeats were made, number of repeats, what the error bars indicate, what data were dropped from the analysis – again, preferably none – etc.) be presented in the figure legends themselves rather than in the Materials and methods to provide immediate context to the readers.

---

## [Author Response]

Essential revisions:1) DBD's marginal stability poses special challenges to any molecular study. The authors must make inferences of the DBD's behavior at 37 ^o^C, while the protein cannot be purified in an unaggregated form at this temperature. Their results are therefore based on measurements taken at 2-22 ^o^C, as noted in the subsection “Extrapolation to physiological conditions”. Even if this is standard practice in the p53 field, I think it's important to note whether this risks any of the conclusions drawn by the authors and provide some reasonable bounds on the errors that this assumption may cause. It seems to me that the very interesting (and major) conclusion of this study that the dGfold of p53 at physiological concentrations may be sensitive to small changes in the concentration of Zinc stands on this analysis. It would therefore be good to assess how safe this conclusion is.

We agree that one of the most far-reaching conclusions of our study is that folding of p53 may be sensitive to small changes in zinc concentration in the cell. This assertion results from ∆G_apo_ being extremely (and unexpectedly) unfavorable at 37 °C, with this figure being extrapolated from measurements taken at lower temperatures. We point out that this method of obtaining thermodynamic folding parameters is commonly used when calorimetry is impractical, and the two techniques have been shown to produce comparable results:

“Introduced by Pace and Laurents in 1989 (Pace and Laurents, 1989), this method has been shown to reproduce ∆C_p_, T_m_, and ∆H_m_ obtained by scanning calorimetry for a number of proteins (Pace et al., 1999; Talla-Singh and Sites, 2008.”

To assess the validity of the ∆G_apo_ extrapolation, we compare our results to those of Prof. Alan Fersht’s group, who carried out a similar analysis of holoDBD stability (Bullock et al., 2000):

“The conclusion that WT p53 folding is sensitive to [Zn^2+^]_free_ fluctuations in the physiological regime carries significant implications, and uncertainty of the ∆G_apo_ extrapolation as well as technical artifacts must be considered. […] Given DBD’s high affinity for zinc (K_Zn_ = 1.6 fM), the only way to obtain ∆G_holo_ = -3.0 kcal mol^-1^ at 37 °C and 2.5 μM total zinc is for apoDBD to be as unstable as the landscape model predicts.”

In other words, if apoDBD possessed a “normal” ∆C_p_, the Fersht group would have measured a much larger, more negative ∆G_fold_. In the original manuscript we noted that ∆C_p_ of apo DBD (7.0 kcal/mol/K) is almost twice as large as that measured for holoDBD by the Fersht group (3.8 kcal/mol/K) (we went on to describe the additional experiments that we performed to verify this anomalously large number). We now correct ourselves in that the ∆C_p_ value in the Fersht study was not measured but calculated, based on the change in solvent accessible surface area between the native structure and a model of the denatured state.

2) In the Materials and methods, the authors note that "outliers were identified on inspection and excluded based on gross divergence from the modeled fit". It would be preferrable to see all of the measurements that were done accurately and see the mean and standard errors that result from all of the data. An alternative is for the authors to provide a detailed explanation on the criteria used to exclude data in each figure. I also strongly recommend that any details on the statistical analysis (what type of repeats were made, number of repeats, what the error bars indicate, what data were dropped from the analysis – again, preferably none – etc.) be presented in the figure legends themselves rather than in the Materials and methods to provide immediate context to the readers.

We now include details of the number and type of replicates, statistical analyses, data exclusion, etc. in the figure legends (with further details in the Materials and methods) as directed:

“Samples sizes were adjusted to provide smooth interpolation across the experimental range; experimental range spanned at least one order of magnitude greater the highest and lower than the smallest central value (e.g. K_d_, C_m_, EC_50_) unless constrained by physical limitations (solubility, buffer range, etc.). […] For multiple outliers, the Holm-Bonferroni correction was applied when calculating the 95% prediction interval. R code for outlier testing is available at https://codeberg.org/AlanBlayney/mutant-p53.”

Re-analyzing the data necessitated three changes to the manuscript.

1) In the original submission we excluded two data points in the Y220C plot (Figure 3—figure supplement 1, red data points). Their rejection was unjustified, and we now include them. This removes the deflection in the curve and causes K_Zn_ to be undetermined. We removed K_Zn_ of Y220C from Figure 3—source data 1 (formerly Figure 4) and deleted the Y220C data point from Figure 3C. This change is consistent with Y220C’s lack of reactivation/refolding by ZMC1, as Y220C now behaves like the other Tyr-to-Cys mutants in Figure 3—figure supplement 2 in that they fail to respond to ZMC1 due to creation of extra zinc coordination sites in the unfolded state. We have added all Tyr-to-Cys mutants in Figure 3—source data 1 to reflect this observation, indicating by an asterisk that K_Zn_ could not be determined, and that they belong either to the pure stability class or mixed zinc-binding/stability class. These changes do not affect any of the other conclusions of the paper.

2) In Figure 8—figure supplement 1B, we now exclude the 1 μM ZMC1 data points from the SUM149PT (p53-R349S) cell survival curve, mark these as rejected, and provide justification in the legend. Refitting the data indicates that SUM149PT cells show a partial response to ZMC1. We change the text from “To confirm the result for M237I, we tested two human cancer cell lines that express p53^M237I^ and found that they too were insensitive to ZMC1 (Figure 8—figure supplement 1B).” to the following:

“To confirm the result for M237I, we tested two human cancer cell lines that express p53^M237I^. One cell line (T98G) was insensitive to ZMC1 and the other (SUM149PT) showed a partial response in which cell viability remained at ~40 % at 10 μM ZMC1; Figure 8—figure supplement 1B).”

The legend to Figure 8—figure supplement 1 now includes:

“The open triangles in the SUM149PT data were excluded from the fit, as they were outside the 95 % prediction interval and remained so when applying the Bonferroni correction for multiple tests.”

The balance of the data indicate that M237I is not refolded or reactivated by ZMC1 and this conclusion remains unchanged in the revised manuscript.

We include the data and statistical analyses that were used to generate the heat map of DNA-binding affinities (Figure 4, formerly Figure 5) as new source data (Figure 4—source data 1).

Finally, applying the exclusion criteria confirmed that we were justified in rejecting the data points in Figure 2C (omitted in the original manuscript). We now show the rejected points.